# Bayesian Basis Function Approximation for Scalable Gaussian Process Priors in Deep Generative Models

**Mehmet Yiğit Balık** [1]  **Maksim Sinelnikov** [1]  **Priscilla Ong** [1]  **Harri Lähdesmäki** [1]

## Abstract

High-dimensional time-series datasets are common in domains such as healthcare and economics. Variational autoencoder (VAE) models, where latent variables are modeled with a Gaussian process (GP) prior, have become a prominent model class to analyze such correlated datasets. However, their applications are challenged by the inherent cubic time complexity that requires specific GP approximation techniques, as well as the general challenge of modeling both shared and individual-specific correlations across time. Though inducing points enhance GP prior VAE scalability, optimizing them remains challenging, especially since discrete covariates resist gradient-based methods. In this work, we propose a scalable basis function approximation technique for GP prior VAEs that mitigates these challenges and results in linear time complexity, with a global parametrization that eliminates the need for amortized variational inference and the associated amortization gap, making it well-suited for conditional generation tasks where accuracy and efficiency are crucial. Empirical evaluations on synthetic and real-world benchmark datasets demonstrate that our approach not only improves scalability and interpretability but also drastically enhances predictive performance.

## 1. Introduction

High-dimensional time-series datasets are common in various domains, including social sciences, economics, and healthcare. These longitudinal datasets contain repeated measurements of subjects over time along with auxiliary covariate information, such as time, age, and gender, in the healthcare domain. By studying the associations between covariates and samples (i.e., measurements), it is possible to model temporal dynamics, such as disease progression, and impute missing values in the data (Diggle, 2002). However, analysis is challenged by factors such as high dimensionality, non-trivial correlations within and across subjects, time-varying covariates, and missing values and observations (Ramchandran et al., 2021).

Several machine learning methods have been developed to address these challenges, with variational autoencoders (VAEs) emerging as an established class of models for representation learning and generative tasks (Kingma & Welling, 2014; Rezende et al., 2014). VAEs can learn both an expressive neural network parameterized model for high-dimensional data generation as well as an efficient amortized variational approximation of sample-specific latent variables. However, standard VAEs assume that each data point is independent and identically distributed (*iid*), which limits their ability to capture correlations between samples in temporal datasets.

VAEs can be enhanced to model correlations between samples. Among these enhancements, VAEs equipped with a Gaussian process (GP) prior for latent variables have proven to be particularly effective for capturing time dependencies and correlations between samples and achieve impressive performance improvements (Casale et al., 2018). However, this powerful approach comes with a cost of computational complexity that scales cubically with the size of the dataset (Quinonero-Candela & Rasmussen, 2005; Bauer et al., 2016) that limits its scalability for large datasets. There have been efforts to tackle these challenges by incorporating inducing points (Titsias, 2009) and stochastic gradient-based optimization with mini-batching (Hensman et al., 2013). Optimizing inducing point locations becomes non-trivial with discrete covariates as their non-continuous kernels preclude gradient-based updates. To our knowledge, no efficient general solution for discrete covariates in GP prior VAEs has been proposed. Finally, their variational posterior commonly uses a mean-field approximation with parameters tied by a shared encoder, which limits performance and introduces an unavoidable amortization gap (Cremer et al., 2018; Margossian & Blei, 2024).

---

[1]Department of Computer Science, Aalto University, Espoo, Finland. Correspondence to: Mehmet Yiğit Balık <mehmet.balik@aalto.fi>.

*Proceedings of the 42$^{nd}$ International Conference on Machine Learning*, Vancouver, Canada. PMLR 267, 2025. Copyright 2025 by the author(s).

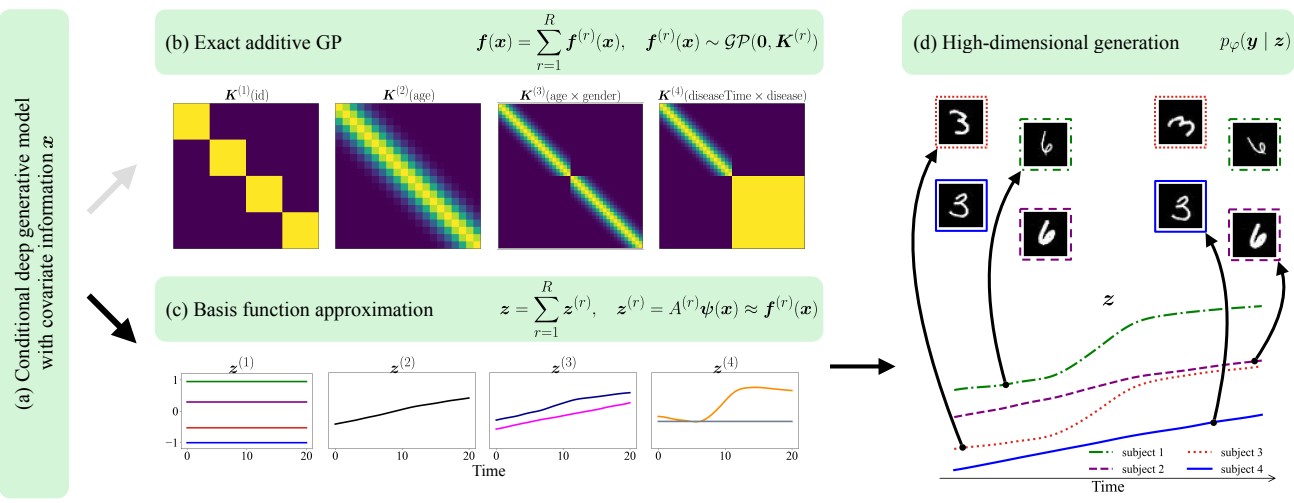

Figure 1. Overview of DGBFGP. (a) A conditional generative model incorporates covariate information. (b) Additive GP priors are used but are not scalable. (c) We replace exact GP with a basis function approximation for scalability. (d) Samples are generated by the decoder.

**Contributions.** In this work, we propose Deep Generative Basis Function Gaussian Process (DGBFGP), a novel conditional generative model. Inspired by earlier VAE approaches (Ramchandran et al., 2021; Ong et al., 2024), DGBFGP is capable of accurately modeling high-dimensional longitudinal data by capturing complex correlations within and across subjects, specified by extensive continuous and categorical covariates using scalable basis function approximation that provides global parameterization and learns parameters using variational inference (VI) (Hoffman et al., 2013). Figure 1 overviews DGBFGP, which uses scalable GP approximations with auxiliary covariates to model individual, group, and temporal effects. The model delivers flexible, interpretable representations and scales to large datasets. Our contributions are:

- We propose DGBFGP, a conditional deep generative model that leverages basis function approximation for mixed-domain additive GPs to model complex correlations within and across subjects, guided by extensive continuous and categorical covariates.

- Our global parameterization avoids explicit kernels, runs in linear time, eliminates the amortization gap, overcomes the limitations of categorical inducing point optimization, enhances interpretability in the latent space by quantifying the contributions of different covariates/effects using Sobol indices, allows for standard mini-batch training, and treats kernel hyperparameters probabilistically.

- We compare DGBFGP against state-of-the-art methods in the literature and report highly competitive performance on synthetic and real-world benchmark datasets.

## 2. Related Work

There has been a vast array of suggestions to improve the performance of VAEs. Sohn et al. (2015) proposed the conditional VAE (CVAE), which leverages auxiliary covariate information to better guide the latent space representation and generate more context-aware outputs. However, it uses a standard Gaussian prior, which is not flexible for complex tasks. The expressiveness of approximated posteriors can be strengthened by importance sampling (Burda et al., 2016), normalizing flows (Rezende & Mohamed, 2015), RealNVPs (Dinh et al., 2017), and inverse autoregressive flows (Kingma et al., 2016). Yet, these methods still rely on the *iid* assumption across samples. Hence, they cannot capture sample correlations.

Casale et al. (2018) proposed the GP prior VAE (GPPVAE), which incorporates GP priors to include view and object information directly into the latent variables. However, its reliance on a restrictive view-object GP product kernel limits its ability to capture subject-specific temporal patterns, making it less suitable for longitudinal studies. Additionally, the pseudo-mini-batch stochastic gradient descent (SGD) training scheme lacks scalability for large datasets as it necessitates a full dataset pass for every training step.

Building upon the idea of employing GPs within the latent space of VAEs, Fortuin et al. (2020) introduced GP-VAE, which assumes a separate GP prior for the time-series data of each individual subject. While GP-VAE is tailored for temporal data, its independent GP priors for each subject limit its ability to capture shared temporal patterns. Furthermore, GP-VAE cannot utilize auxiliary covariate information beyond time, restricting its flexibility in context-rich

*Table 1.* Features of our proposed method contrasted to the related previous methods.

| MODEL | SHARED TEMPORAL STRUCTURE | INDIVIDUAL TEMPORAL STRUCTURE | OTHER COVARIATES | MINIBATCHING | GENERATIVE | GP COMPLEXITY | REFERENCE |
|---|---|---|---|---|---|---|---|
| CVAE | ✓ | ✗ | ✓ | ✓ | ✓ | - | SOHN ET AL. (2015) |
| BRITS | ✓ | ✗ | ✗ | ✓ | ✗ | - | CAO ET AL. (2018) |
| L-NODE | ✓ | ✓ | ✗ | ✓ | ✓ | - | IAKOVLEV ET AL. (2023) |
| GPPVAE | ✓ | LIMITED | LIMITED | PSEUDO | ✓ | $\mathcal{O}(NH^2)$ | CASALE ET AL. (2018) |
| GP-VAE | ✗ | ✓ | ✗ | ✓ | ✓ | $\mathcal{O}(\sum_p n_p^3)$ | FORTUIN ET AL. (2020) |
| SVGP-VAE | ✓ | LIMITED | LIMITED | ✓ | ✓ | $\mathcal{O}(NM^2 + M^3)$ | JAZBEC ET AL. (2021) |
| LVAE | ✓ | ✓ | ✓ | ✓ | ✓ | $\mathcal{O}(NM^2 + \sum_p n_p^3)$ | RAMCHANDRAN ET AL. (2021) |
| SGP-BAE | ✓ | LIMITED | ✓ | ✓ | ✓ | $\mathcal{O}(NM^2 + M^3)$ | TRAN ET AL. (2023) |
| DGBFGP | ✓ | ✓ | ✓ | ✓ | ✓ | $\mathcal{O}(N \sum_r B^{(r)})$ | OUR WORK |

datasets. To overcome some of these issues, both SVGP-VAE (Jazbec et al., 2021) and LVAE (Ramchandran et al., 2021) were introduced, leveraging inducing points to enable scalable, mini-batch training (Titsias, 2009; Hensman et al., 2013). While SVGP-VAE inherits GPPVAE's limitations in incorporating all auxiliary covariates, LVAE is capable of handling an arbitrary number of covariates, allowing greater flexibility in modeling diverse datasets. However, both approaches face challenges in initializing and optimizing the inducing points, which can impact their performance and applicability, particularly in complex scenarios. Finally, Tran et al. (2023) proposed SGP-BAE, a fully Bayesian autoencoder using stochastic gradient Hamiltonian Monte Carlo to jointly sample decoder and sparse GP prior parameters. Howver, it still faces the same challenge with categorical covariates due to its reliance on inducing points.

Multi-output GPs extend traditional GPs to model correlations across multiple output dimensions (Alvarez et al., 2012). By learning dependencies both within and across variables, they offer a natural and flexible way to represent coupled latent spaces. This capability makes them particularly suitable for tasks where multiple related outputs need to be jointly modeled. However, multi-output GPs do not directly scale to high-dimensional datasets.

Ong et al. (2024) introduced LMM-VAE, which uses linear mixed models (LMMs) as conditional priors in VAEs, enabling the integration of auxiliary covariates while modeling shared and random effects. By combining the strengths of LMMs and VAEs, it captures nuanced dependencies and is adaptable to diverse applications. Its performance can be further improved with basis functions for greater flexibility in modeling complex patterns.

Modeling temporal correlations is also explored through latent neural ordinary differential equations (ODEs) (Rubanova et al., 2019). Iakovlev et al. (2023) proposed L-NODE, a latent neural ODE with Bayesian multiple shooting that offers an efficient framework for continuous-time latent dynamics with sparse observations. In contrast, classical approaches like BRITS (Cao et al., 2018) use bidirectional recurrent networks for time-series imputation but are not generative models. These methods address comple-

mentary aspects of time-series analysis and offer diverse tools for high-dimensional datasets. Table 1 summarizes the capabilities of the related previous methods.

## 3. Background

**Problem setup.** We consider general *longitudinal experimental designs* with $P$ unique instances (i.e. subjects or patients), where each instance $p$ has $n_p$ longitudinal samples. The total number of samples equals the sum of instance-specific samples, $N = \sum_{p=1}^{P} n_p$. We denote the high-dimensional samples as $\boldsymbol{y} \in \mathcal{Y} = \mathbb{R}^D$. Without loss of generality, we can assume that the samples are ordered according to the instances such that the first $n_1$ samples $\boldsymbol{y}_1, \ldots, \boldsymbol{y}_{n_1}$ are from instance one, the next $n_2$ samples $\boldsymbol{y}_{n_1+1}, \ldots, \boldsymbol{y}_{n_1+n_2}$ from instance two, etc. Samples across all instances are denoted as $Y = [\boldsymbol{y}_1, \ldots, \boldsymbol{y}_N]$. Additionally, an instance $p$ has corresponding covariate information for each sample, collectively denoted as $X = [\boldsymbol{x}_1, \ldots, \boldsymbol{x}_N]$, where $\boldsymbol{x} \in \mathcal{X} = \prod_{q=1}^{Q} \mathcal{X}_q$ is a $Q$-dimensional vector including information such as time, age and gender. We denote the sample-specific $L$-dimensional ($L \ll D$) latent variables as $Z = [\boldsymbol{z}_1, \ldots, \boldsymbol{z}_N]$, where each $\boldsymbol{z} \in \mathcal{Z} = \mathbb{R}^L$.

### 3.1. Variational autoencoders

Consider a data sample $\boldsymbol{y}$ generated by a joint generative model $p_\omega(\boldsymbol{y}, \boldsymbol{z}) = p_\varphi(\boldsymbol{y}|\boldsymbol{z})p_\theta(\boldsymbol{z})$, where $\omega = \{\varphi, \theta\}$ denotes parameters. The posterior $p_\omega(\boldsymbol{z}|\boldsymbol{y}) = p_\varphi(\boldsymbol{y}|\boldsymbol{z})p_\theta(\boldsymbol{z})/p_\omega(\boldsymbol{y})$ is generally intractable since the term $p_\omega(\boldsymbol{y})$ cannot be computed analytically. A common approach is to employ amortized VI (AVI) (Kingma & Welling, 2014; Rezende et al., 2014) to estimate the true posterior with a parameterized approximation $q_\phi(\boldsymbol{z}|\boldsymbol{y})$. This is done by optimizing a lower bound on the evidence, $\log p_\omega(\boldsymbol{y})$, w.r.t. all parameters involved

$$\log p_\omega(\boldsymbol{y}) \geq \mathbb{E}_{q_\phi}\left[\log p_\varphi(\boldsymbol{y} \mid \boldsymbol{z})\right] - \mathrm{KL}(q_\phi(\boldsymbol{z} \mid \boldsymbol{y}) \,\|\, p_\theta(\boldsymbol{z})).$$

A common assumption is that the likelihood $p_\varphi(\boldsymbol{y} \mid \boldsymbol{z})$, the prior $p_\theta(\boldsymbol{z})$, and the variational posterior $q_\phi(\boldsymbol{z} \mid \boldsymbol{y})$ are all represented using a mean-field approximation. In practice, this implies that distributions factorize over samples.

Sohn et al. (2015) build upon this idea by incorporating auxiliary covariates $\boldsymbol{x} \in \mathcal{X}$ into the encoder and decoder networks. This allows the VI procedure to approximate not just $p_\omega(\boldsymbol{z}|\boldsymbol{y})$, but $p_\omega(\boldsymbol{z}|\boldsymbol{y}, \boldsymbol{x})$ for the conditional generative model $p_\omega(\boldsymbol{y}, \boldsymbol{z}|\boldsymbol{x}) = p_\varphi(\boldsymbol{y}|\boldsymbol{z}, \boldsymbol{x})p_\theta(\boldsymbol{z}|\boldsymbol{x})$. Although CVAE enables more flexible generative modeling by leveraging known factors, it shares the *iid* assumption of the VAE.

### 3.2. Gaussian processes

GPs extend the concept of the Gaussian probability distribution of random variables to stochastic processes, which characterize the properties of entire functions (Rasmussen & Williams, 2006). A univariate GP

$$f(\boldsymbol{x}) \sim \mathcal{GP}(\mu(\boldsymbol{x}), k(\boldsymbol{x}, \boldsymbol{x}'))$$

is fully determined by its mean function $\mu(\boldsymbol{x})$ (generally considered as 0) and covariance (i.e. kernel) function $k(\boldsymbol{x}, \boldsymbol{x}')$. For a finite collection of inputs $X = [\boldsymbol{x}_1, \ldots, \boldsymbol{x}_N]$, GP evaluated at $X$, $f(X) = [f(\boldsymbol{x}_1), \ldots, f(\boldsymbol{x}_N)]^\top$, is defined to have a joint Gaussian distribution $f(X) \sim \mathcal{N}(\boldsymbol{0}, \boldsymbol{\Sigma})$, where $\boldsymbol{\Sigma}_{i,j} = k(\boldsymbol{x}_i, \boldsymbol{x}_j)$. The covariance function encodes our prior assumptions about the underlying latent functions that generate the data. For example, a kernel for continuous-valued covariate might assume that the function changes smoothly (ensuring continuity), whereas a kernel for categorical covariate defines properties of the function across the categories. These covariance functions generally have hyperparameters, which are inferred from data.

### 3.3. Gaussian process prior variational autoencoders

A GP conditioned on auxiliary information can capture nontrivial dependencies and provide a flexible, non-parametric model that can adapt to complex patterns in the data. Previously, researchers have utilized GPs as priors in VAEs to create effective generative models (Casale et al., 2018).

Let $\boldsymbol{f} : \mathcal{X} \rightarrow \mathcal{Z}$ and $\boldsymbol{z} = \boldsymbol{f}(\boldsymbol{x}) = [f_1(\boldsymbol{x}), \ldots, f_L(\boldsymbol{x})]^\top$, and assume that $\boldsymbol{f}$ follows a multi-output GP prior (Alvarez et al., 2012):

$$\boldsymbol{f}(\boldsymbol{x}) \sim \mathcal{GP}(\boldsymbol{0}, \boldsymbol{K}(\boldsymbol{x}, \boldsymbol{x}' \mid \theta)),$$

where $\boldsymbol{K}(\boldsymbol{x}, \boldsymbol{x}' \mid \theta)$ is a so-called cross-covariance function with hyperparameters $\theta$. Contrary to standard multi-output GPs that typically use a linear model of co-regionalization to define the cross-covariance function (Alvarez et al., 2012), GP prior VAE models assume that each latent dimension has an independent GP prior, i.e., factorizes across dimensions

$$p(\boldsymbol{f}(X) \mid \theta) = \prod_{l=1}^{L} \mathcal{N}(\boldsymbol{0}, \boldsymbol{\Sigma}_l), \tag{1}$$

where $\boldsymbol{\Sigma}_l$ denotes the covariance matrix of the $l^{\text{th}}$ component of $\boldsymbol{f}(\cdot)$. This assumption comes without any loss of

generality because the latent variables are further mapped to the data space by a neural network parameterized decoder $p_\varphi(\boldsymbol{y} \mid \boldsymbol{z})$ that can introduce arbitrary correlations at least as expressive as standard multi-output GPs across dimensions.

Motivated by longitudinal datasets, Ramchandran et al. (2021) proposed to use an additive GP prior for each of the latent dimensions:

$$f_l(\boldsymbol{x}) = f_l^{(1)}(\boldsymbol{x}^{(1)}) + \cdots + f_l^{(R)}(\boldsymbol{x}^{(R)})$$
$$f_l^{(r)}(\boldsymbol{x}^{(r)}) \sim \mathcal{GP}\left(0, k_l^{(r)}\big(\boldsymbol{x}^{(r)}, \boldsymbol{x}^{(r)\prime} \mid \theta_l^{(r)}\big)\right),$$

where $R$ denotes the number of additive components, each component depends on either a single or a pair of covariates $\boldsymbol{x}^{(r)} \in \mathcal{X}^{(r)} \subseteq \mathcal{X}$, and $\theta_l^{(r)}$ denotes the kernel hyperparameters of the $r^{\text{th}}$ component of the $l^{\text{th}}$ latent dimension. This additive GP model provides interpretability similar to commonly used LMMs (Laird & Ware, 1982) do and allows for handling mixed-domain (e.g. a combination of continuous and categorical) inputs. The additive GP model implies that the factorizable joint distribution $p(\boldsymbol{f}(X) \mid \theta)$ from Equation (1) has an additive covariance structure $\boldsymbol{\Sigma}_l = \sum_{r=1}^{R} \boldsymbol{\Sigma}_l^{(r)}$, where $i, j$ element of the covariance matrix $\boldsymbol{\Sigma}_l^{(r)}$ is given by $k_l^{(r)}(\boldsymbol{x}_i^{(r)}, \boldsymbol{x}_j^{(r)} \mid \theta_l^{(r)})$.

Being able to model correlations comes with a chain of trade-offs. We need to keep a kernel matrix with $\mathcal{O}(N^2)$ memory complexity. In addition, $\mathcal{O}(N^3)$ time complexity arises from the inversion of the kernel matrix, necessitating reliable approximative GP techniques. Previous GP prior VAEs have proposed to use the inducing point approach (Ramchandran et al., 2021; Jazbec et al., 2021) achieving $\mathcal{O}(NM^2 + M^3)$ time complexity with $M$ inducing points, or low-rank approximations (Casale et al., 2018) reducing complexity to $\mathcal{O}(NH^2)$, where $H \ll N$ is the dimensionality of the low-rank approximation. Instead, in this work, we propose an efficient method that uses basis function approximation, which results in global parametrization.

## 4. Methods

### 4.1. Kernel approximation for continuous covariates

Stationary GP kernels, such as the squared exponential (SE) kernel $k_{\text{se}}(x, x'|\sigma, \ell) = \sigma^2 \exp(-\frac{\|x-x'\|^2}{2\ell^2})$, can be written as a function of the distance between the two inputs, $k(r) = k(x, x') = k(|x - x'|)$, where $x \in \mathbb{R}$ is a univariate continuous input covariate (e.g. time, age, etc.). According to Bochner's theorem (Akhiezer & Glazman, 2013; Rasmussen & Williams, 2006), covariance functions can be represented as the Fourier transform of a positive measure since they are positive definite. When a measure has an associated density, it is referred to as the spectral density $s(\omega)$ of the covariance function. This relationship leads to the Fourier duality between spectral densities and kernels, a

principle known as the Wiener-Khintchin theorem:

$$s(\omega) = \mathcal{F}\{k(r)\} = \int_{-\infty}^{\infty} k(r)e^{-i\omega r} dr$$

$$k(r) = \mathcal{F}^{-1}\{s(\omega)\} = \frac{1}{2\pi} \int_{-\infty}^{\infty} s(\omega)e^{i\omega r} d\omega.$$

For example, the spectral density corresponding to the SE kernel is given by $s_{\text{se}}(\omega \mid \sigma, \ell) = \sigma^2 \ell \sqrt{2\pi} \exp(-\frac{\ell^2 \omega^2}{2})$ which is Gaussian in the frequency domain.

Solin & Särkkä (2020) demonstrated that an isotropic covariance function $(k(r) \triangleq k(||r||))$ has the following eigenfunction approximation

$$k(x, x') \approx \tilde{k}(x, x') = \sum_{m=1}^{M} s(\sqrt{\lambda_m})\phi_m(x)\phi_m(x'), \quad (2)$$

where $\lambda_m$ and $\phi_m$ are the $m^{\text{th}}$ eigenvalue and eigenfunction of the Laplace operator with Dirichlet boundary conditions, including boundary $\Omega = [-J, J] \in \mathbb{R}$ described as

$$\begin{cases} -\frac{\partial^2 \phi_m(x)}{\partial x^2} = \lambda_m \phi_m(x), & x \in \Omega \\ \phi_m(x) = 0, & x \in \partial\Omega. \end{cases}$$

In this setup with $J > 0$, the eigenfunctions and eigenvalues are $\phi_m(x) = \frac{1}{\sqrt{J}}\sin(\frac{\pi m(x+J)}{2J})$ and $\lambda_m = (\frac{\pi m}{2J})^2$, respectively. Finally, we can represent Hilbert space approximation of a GP in a linear parametric form with $\phi(x) = [\phi_1(x), \ldots, \phi_M(x)]^\top$ and global parameters $\boldsymbol{a} \in \mathbb{R}^M$ as

$$f(x) \approx \boldsymbol{a}^\top \phi(x), \quad (3)$$

where $\boldsymbol{a} \sim \mathcal{N}(\boldsymbol{0}, \boldsymbol{S})$ and $\boldsymbol{S} = \text{diag}(s(\sqrt{\lambda_1}), \ldots, s(\sqrt{\lambda_M}))$. Note that the approximation depends on the kernel hyperparameters via the parameter prior, which for the SE kernel can be written as $\boldsymbol{a} \sim \mathcal{N}(\boldsymbol{0}, \boldsymbol{S}(\sigma, \ell))$ and $\boldsymbol{S}(\sigma, \ell) = \text{diag}(s_{\text{se}}(\sqrt{\lambda_1} \mid \sigma, \ell), \ldots, s_{\text{se}}(\sqrt{\lambda_M} \mid \sigma, \ell))$.

For the remainder of this paper, we will use the SE kernel to model continuous covariates. However, our method is applicable to any stationary continuous covariance function.

### 4.2. Generative model

Ong et al. (2024) show that a Fourier basis-based linear model can be utilized to approximate the latent space of GP prior VAEs. Here we show how an additive GP prior VAE model can be defined using the Hilbert space approximation and the associated global parametrization. Let $\boldsymbol{x}^{(r)} \in \mathbb{R}$ denote one of the continuous covariates from $\mathcal{X}$ as before. To simplify notation, we assume that all additive components use the same number of eigenfunctions, $M$, for the kernel approximation in Equation (2), but that can be generalized easily. Assuming the latent dimensions are a priori independent as in previous works (see e.g. Equation (1)), we can

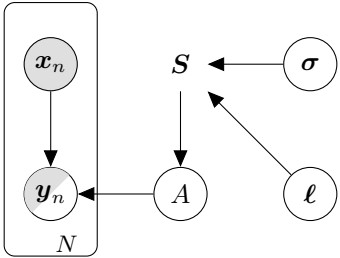

*Figure 2.* Graphical model of DGBFGP. Blank, partially shaded, and shaded circles indicate latent, partially observed, and observed variables, respectively. Non-circled variable $\boldsymbol{S}$ is deterministic.

obtain the Hilbert space approximation for the $r^{\text{th}}$ component of a multi-output GP prior by using the approximation from Equation (3) for each latent dimension separately

$$\boldsymbol{z}^{(r)} = A^{(r)}\phi(\boldsymbol{x}^{(r)}) = \begin{bmatrix} \boldsymbol{a}_{r1}^\top \\ \vdots \\ \boldsymbol{a}_{rL}^\top \end{bmatrix} \phi(\boldsymbol{x}^{(r)}),$$

where $\boldsymbol{a}_{rl} \in \mathbb{R}^M$ denotes the linear model parameters for additive component $r$ and latent dimension $l$, $\phi(\boldsymbol{x}^{(r)}) \in \mathbb{R}^M$, and $\boldsymbol{a}_{rl} \sim \mathcal{N}(\boldsymbol{0}, \boldsymbol{S}_r(\sigma_r, \ell_r))$. Here we assume that kernel hyperparameters are shared across dimensions although this can be easily generalized, and we explicitly write the prior dependence of $\boldsymbol{a}_{rl}$ on the corresponding kernel hyperparameters $\sigma_r$ and $\ell_r$ via the spectral density. Since GP prior factorizes across latent dimensions, the prior on $A^{(r)}$ is defined as $p(A^{(r)}) = \prod_{l=1}^{L} p(\boldsymbol{a}_{rl})$. Therefore, the full additive GP structure is approximated by

$$\boldsymbol{z} = \sum_{r=1}^{R} \boldsymbol{z}^{(r)} = \sum_{r=1}^{R} A^{(r)}\phi(\boldsymbol{x}^{(r)})$$

which is parameterized by a collection of weights $A = (A^{(1)}, \ldots, A^{(R)})$ with prior $p(A) = \prod_{r=1}^{R} p(A^{(r)})$. See Appendix B for details. The generative model can then be expressed as follows (see Figure 2 for the plate diagram):

$$\sigma_r \sim \text{Lognormal}(0, 1) \quad (r = 1, \ldots, R)$$

$$\ell_r \sim \text{Lognormal}(0, 1)$$

$$\boldsymbol{S}_r(\sigma_r, \ell_r) = \text{diag}\big(\{s_{\text{se}}(\sqrt{\lambda_m} \mid \sigma_r, \ell_r\}_{m=1}^{M}\big)$$

$$A \mid \boldsymbol{\sigma}, \boldsymbol{\ell} \sim \prod_{r=1}^{R} \prod_{l=1}^{L} \mathcal{N}(\boldsymbol{a}_{rl} \mid \boldsymbol{0}, \boldsymbol{S}_r(\sigma_r, \ell_r)) \quad (4)$$

$$Y \mid A, X \sim \prod_{n=1}^{N} p_\varphi(\boldsymbol{y}_n \mid A, \boldsymbol{x}_n),$$

where $\boldsymbol{\sigma} = (\sigma_1, \ldots, \sigma_R)$, $\boldsymbol{\ell} = (\ell_1, \ldots, \ell_R)$, and $s_{\text{se}}(\sqrt{\lambda_m} \mid \sigma_r, \ell_r)$ is again the spectral density of the SE kernel that depends on the kernel hyperparameters $\sigma_r$ and $\ell_r$ evaluated

at the square-root of the eigenvalue $\lambda_m$. A log-normal prior is chosen for both the length scale $\ell_r$ (Timonen et al., 2021) and amplitude $\sigma_r$ as it ensures positivity and accommodates a wide range of values. The likelihood function $p_\varphi$ is parameterized by a neural network.

### 4.3. Interpretable model for mixed-domain covariates

We have so far described a conditional deep generative model with a highly scalable GP prior for continuous covariates. But it is also possible to model categorical covariates and kernels (Garrido-Merchán & Hernández-Lobato, 2020) utilizing their eigendecompositions. Assume that one of the covariates, $x \in \mathcal{C} = \{1, \ldots, C\}$, has $C$ possible categories, and the corresponding kernel is $k : \mathcal{C} \times \mathcal{C} \to \mathbb{V} \subset \mathbb{R}$. Let $C$ denote the $C \times C$ kernel matrix. Symmetric kernels have the eigendecomposition $C = \Theta D \Theta^\top$, where $D$ is a diagonal matrix containing $C$ eigenvalues on the diagonal and $\Theta$ has $C$ eigenvectors as its columns. Following Timonen & Lähdesmäki (2024), by defining function $\vartheta_c : \mathcal{C} \to \mathbb{R}$ that outputs the values of the $c^{\text{th}}$ column of $\Theta$, the covariance function $k$ can be written as

$$
k(x, x') = C_{x,x'} = [\Theta D \Theta^\top]_{x,x'} = \sum_{c=1}^{C} d_c \Theta_{x,c} \Theta_{x',c}
$$
$$
= \sum_{c=1}^{C} d_c \vartheta_c(x) \vartheta_c(x').
$$

The corresponding GP has the parametric form $f(x) = \vartheta(x)a$, where $\vartheta(x)$ denotes the $x^{\text{th}}$ row of $\Theta$, and $a \sim \mathcal{N}(0, D)$. It is important to note that $f(x)$ is the exact GP, not an approximation. Hence, it can be used to model categorical covariates, such as gender and subject IDs. Generalization for new subjects can be achieved by learning only subject-specific parameters (see Appendix B.1).

We are typically interested in modeling joint effects of continuous and categorical covariates (Saves et al., 2023). In our setting, we can compute the interactions between the approximation for the stationary kernel and the decomposition of the categorical kernel (Timonen & Lähdesmäki, 2024):

$$
\tilde{k}(\boldsymbol{x}, \boldsymbol{x'}) = \tilde{k}_{\text{se}}(\boldsymbol{x}, \boldsymbol{x'}) k_{\text{ca}}(\boldsymbol{x}, \boldsymbol{x'})
$$
$$
= \sum_{m=1}^{M} \sum_{c=1}^{C} s(\sqrt{\lambda_m}) d_c \phi_m(\boldsymbol{x}) \phi_m(\boldsymbol{x'}) \vartheta_c(\boldsymbol{x}) \vartheta_c(\boldsymbol{x'}) \quad (5)
$$

where $\tilde{k}_{\text{se}}(\boldsymbol{x}, \boldsymbol{x'})$ depends only on one continuous variable and $k_{\text{ca}}(\boldsymbol{x}, \boldsymbol{x'})$ depends only on one categorical variable of inputs. Defining $B = MC$, the product kernel can then be expressed as $\tilde{k}(\boldsymbol{x}, \boldsymbol{x'}) = \sum_{b=1}^{B} s_b \psi_b(\boldsymbol{x}) \psi_b(\boldsymbol{x'})$, where

$$
\boldsymbol{\psi}(\boldsymbol{x}) = \begin{bmatrix} \phi_1(\boldsymbol{x})\vartheta_1(\boldsymbol{x}) \\ \phi_1(\boldsymbol{x})\vartheta_2(\boldsymbol{x}) \\ \vdots \\ \phi_M(\boldsymbol{x})\vartheta_C(\boldsymbol{x}) \end{bmatrix} \quad \text{and} \quad \boldsymbol{s} = \begin{bmatrix} s(\sqrt{\lambda_1})d_1 \\ s(\sqrt{\lambda_1})d_2 \\ \vdots \\ s(\sqrt{\lambda_M})d_C \end{bmatrix} \quad (6)
$$

are both $B$ dimensional. The GP with interaction kernel given in Equation (5) has the following parametric form

$$
f(\boldsymbol{x}) \approx \boldsymbol{a}^\top \boldsymbol{\psi}(\boldsymbol{x}), \quad \boldsymbol{a} \sim \mathcal{N}(\boldsymbol{0}, \text{diag}(\boldsymbol{s})).
$$

We showed in Sections 4.1-4.2 that $\boldsymbol{a}_{rl}$, the global weights for continuous variables, have a multivariate Gaussian prior with diagonal covariance in Equation (4). This structure extends directly to the interaction kernel by incorporating a diagonal covariance with entries that combine the SE kernel's spectral density and the eigenvalues of the categorical kernel. The basis functions are products of the continuous and categorical kernel bases as shown in Equation (6).

### 4.4. Inference

We use VI (Hoffman et al., 2013) to learn the parameters of the generative model by assuming a factorized variational posterior, $q(A, \boldsymbol{\sigma}, \boldsymbol{\ell}) = q(A)q(\boldsymbol{\sigma})q(\boldsymbol{\ell})$. Approximate posterior $q(A)$ is Gaussian whereas $q(\boldsymbol{\sigma})$ and $q(\boldsymbol{\ell})$ are log-normal. The ELBO is given as (see Appendix C for the derivation)

$$
\log p(Y \mid X) \geq \mathbb{E}_{q(A)} \left[ \sum_{n=1}^{N} \log p_\varphi(\boldsymbol{y}_n \mid A, \boldsymbol{x}_n) \right]
$$
$$
- \mathbb{E}_{q(\boldsymbol{\sigma})q(\boldsymbol{\ell})}[\text{KL}(q(A) \,\|\, p(A \mid \boldsymbol{\sigma}, \boldsymbol{\ell}))] \quad (7)
$$
$$
- \text{KL}(q(\boldsymbol{\sigma}) \,\|\, p(\boldsymbol{\sigma}))
$$
$$
- \text{KL}(q(\boldsymbol{\ell}) \,\|\, p(\boldsymbol{\ell})),
$$

where we use Monte Carlo to sample $\boldsymbol{\sigma}$ and $\boldsymbol{\ell}$ from the corresponding distributions $q(\boldsymbol{\sigma})$ and $q(\boldsymbol{\ell})$ in order to estimate $\mathbb{E}_{q(\boldsymbol{\sigma})q(\boldsymbol{\ell})}[\text{KL}(q(A) \,\|\, p(A \mid \boldsymbol{\sigma}, \boldsymbol{\ell}))]$. These values are then substituted into the spectral density computation. Note that the spectral densities are needed to define $p(A \mid \boldsymbol{\sigma}, \boldsymbol{\ell})$ as shown in Equation (4). The closed form of this KL term is given in Appendix C, Equation (11), which offers a simpler solution compared to the more complex formulations involving GPs in previous GP prior VAE models. Additionally, the KL terms for $\boldsymbol{\sigma}$ and $\boldsymbol{\ell}$ are provided in Equation (12). Since the parameterization is global, the ELBO (7) is directly amenable to SGD optimization with mini-batching.

We also present a semi-amortized version of our model, denoted as SA-DGBFGP in Appendix E, where AVI is used only for modeling instance-specific characteristics.

### 4.5. Computational complexity

The Hilbert space approximation admits a closed-form eigendecomposition independent of specific kernels, with hyperparameters entering only via spectral densities that also have closed-form solutions, enabling $\mathcal{O}(1)$ plug-and-play use of eigenvalues and eigenfunctions. Computing the categorical kernel eigendecomposition once suffices, so overhead is negligible as $C \ll N$. Hence, the most computationally demanding operation in DGBFGP model is

the multiplication of each data point's basis function representation, $\phi(\boldsymbol{x}_n^{(r)}) \in \mathbb{R}^B$, with the global linear model parameters, $\boldsymbol{a}_{rl} \in \mathbb{R}^B$. This computation is performed for $N$ data points. Extending this operation across all components while accounting for the possibility that each component may have a different number of basis functions, the total time complexity for a single latent dimension becomes $\mathcal{O}(N \sum_{r=1}^{R} B^{(r)})$.

### 4.6. Sobol indices for quantifying interpretability

In additive GPs, we can use a sensitivity analysis based on Sobol indices (Sobol, 1993) to quantify the relative contribution of additive components (Lu et al., 2022). The Sobol index for $r^{\text{th}}$ component is defined as $\text{SI}^{(r)} = \frac{\mathbb{V}[f^{(r)}(\boldsymbol{x}^{(r)})]}{\mathbb{V}[\sum_{i=1}^{R} f^{(i)}(\boldsymbol{x}^{(i)})]}$, which is the fraction of the total GP output variance attributable to the $r^{\text{th}}$ component, thereby offering a clear metric for assessing its impact. It is important to note that in this context, the analysis is focused on the contributions of the model components, not directly on individual covariates. A large $\text{SI}^{(r)}$ signals that component $r$ has a significant contribution, while a small value implies a minor role.

## 5. Experiments

We showcase the effectiveness of our model in temporal interpolation and long-term future prediction through experiments on common benchmarks that include synthetic and real-world longitudinal healthcare datasets. To evaluate our approach, we compare it against state-of-the-art GP prior VAEs, including SVGP-VAE (Jazbec et al., 2021), which improves the scalability of GP prior VAEs using inducing points, LVAE (Ramchandran et al., 2021), which uses inducing points and is specifically designed for longitudinal datasets, as well as SGP-BAE (Tran et al., 2023), which also uses inducing points but treats all parameters in a fully Bayesian manner. We also include L-NODE (Iakovlev et al., 2023) in one experiment with shared governing temporal dynamics across samples, but exclude it from experiments with varying group-specific dynamics. Finally, we also included BRITS (Cao et al., 2018), which implements a bidirectional RNN. In all experiments, we used similar encoder and decoder architectures for all models. We assess predictive performance using mean squared error (MSE), reporting the average and standard deviation over five iterations. In all experiments, we used a fixed number of basis functions, $M = 10$, to approximate continuous kernels. Details on hyperparameter selection and latent space modeling for each experiment are provided in Appendix F, while neural network architectures are described in Appendix G. The implementation of our model is provided at https://github.com/YigitBalik/DGBFGP.

*Table 2.* Temporal interpolation MSE on Rotated MNIST test data.

| Method | Latent dimension | MSE |
|---|---|---|
| BRITS (Cao et al., 2018) | - | $0.989 \pm 0.0098$ |
| SVGP-VAE (Jazbec et al., 2021) | 16 | $0.028 \pm 0.0005$ |
| LVAE (Ramchandran et al., 2021) | 16 | $0.026 \pm 0.0006$ |
| SGP-BAE (Tran et al., 2023) | 16 | $0.023 \pm 0.0006$ |
| L-NODE (Iakovlev et al., 2023) | 16 | $0.017 \pm 0.0002$ |
| DGBFGP (our work) | 16 | $\mathbf{0.009 \pm 0.0001}$ |

### 5.1. Rotated MNIST

We evaluate our model on the temporal interpolation task using a modified version of the handwritten MNIST digits dataset (LeCun et al., 1998). This dataset consists of $P = 400$ unique instances of the digit '3', each rotated across 16 evenly spaced angles covering a full rotation, $[0, 2\pi]$. Each data point is characterized by a unique instance $id$ and an associated rotation $angle$. Temporal interpolation in this context refers to predicting intermediate representations of an instance at unseen angles. See Appendix F.1 for additional details.

Table 2 provides a comparative analysis of our method against existing approaches. Our model achieves a significantly lower MSE on the interpolation task, outperforming previous baselines by a substantial margin in addition to being computationally more efficient. Additionally, Figure 6 in Appendix H presents conditionally generated samples for three test instances, illustrating the ability of our model to capture smooth temporal transitions in the latent space.

In Figure 8, model interpretability is highlighted by examples of latent functions clearly decomposed into two additive components. Normalized Sobol indices show that both components contribute nearly equally, indicating the model effectively captures individual nuances and rotational dynamics. See Figure 9 (Appendix H) for t-SNE visualization.

### 5.2. Health MNIST

To better approximate real-world medical data, we introduce modifications to the MNIST digits dataset simulating a high-dimensional longitudinal dataset (Krishnan et al., 2015) with missing values at each time point. In this setup, the digits '3' and '6' represent two distinct biological genders, with an equal distribution across both groups. A shared aging effect is modeled by gradually shifting all instances toward the bottom right corner over time. Additionally, we assume that half of the subjects in each gender group are healthy, while the other half are unhealthy. The simulated measurements remain unchanged for healthy subjects. Whereas, for unhealthy individuals, rotation is applied where the amount of rotation depends on the time to disease diagnosis. To further mimic noisy real-world measurements, we introduce a random rotational jitter to each data point. Finally, 25% of each sample's pixels are randomly selected and marked

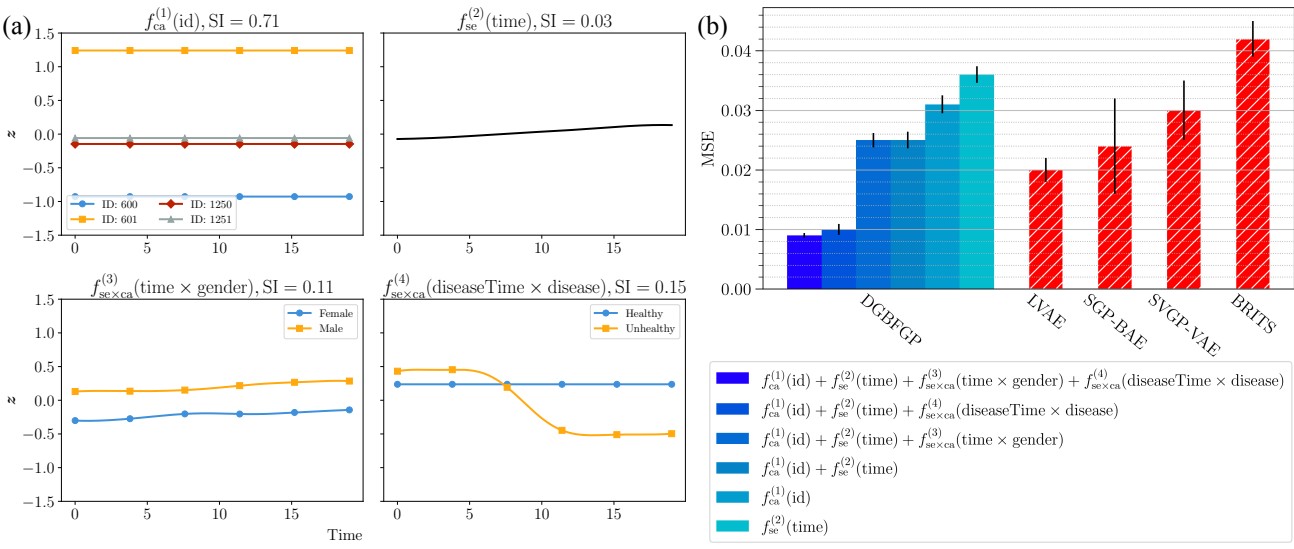

Figure 3. Health MNIST experiment results. (a) Example learned latent functions in the Health MNIST experiment along with normalized Sobol indices averaged over $L = 32$ dimensions. (b) Predictive MSE of different models with 32 latent dimensions on test samples.

as missing. See Figure 10 for illustration. As a result, the dataset includes $Q = 5$ auxiliary covariates, namely $age$, $diseaseTime$, $gender$, and $disease$ along with a subject-specific identifier $id$. The dataset consists of $P = 1,300$ unique subjects, each observed over $n_p = 20$ time points.

For the training set, we include all time points from 1,100 subjects, along with the first five time points from the remaining subjects. The validation and test sets each contains 100 unique subjects, with their remaining time points used for evaluating the model in the future prediction task.

Figure 3(b) shows performance on the future prediction task, where our approach outperforms others. Figure 3(b) presents an ablation removing additive components to show that the numerical value $R$ alone does not determine expressiveness: the design of each additive component, especially the time related components, is essential for model performance even though Figure 3(a) indicates that $id$ is the most dominant Sobol index component. Figure 10 shows visual comparisons for two test subjects, highlighting DGBFGP's ability to capture subject-specific traits and temporal dynamics accurately. Figure 11 in Appendix H depicts the relationship between $M$ and the predictive performance and shows that our approach can achieve competitive performance even with a minimal number of basis functions.

### 5.3. Physionet

We also utilize healthcare data from the Physionet Challenge 2012 (Silva et al., 2012) to benchmark our model's long-term predictive performance in a real-world setting. The dataset comprises approximately 12,000 subjects monitored

Table 3. Predictive MSE on test subjects of the Physionet dataset.

| Method | Latent dimension | MSE |
|---|---|---|
| Mean prediction (all training samples) | - | $0.904 \pm 0.000$ |
| Mean prediction (training samples of test subjects) | - | $0.756 \pm 0.000$ |
| SVGP-VAE (Jazbec et al., 2021) | 32 | $0.802 \pm 0.009$ |
| SGP-BAE (Tran et al., 2023) | 32 | $0.781 \pm 0.002$ |
| BRITS (Cao et al., 2018) | - | $0.732 \pm 0.005$ |
| LVAE (Ramchandran et al., 2021) | 32 | $0.718 \pm 0.007$ |
| DGBFGP (our work) | 32 | $\mathbf{0.619 \pm 0.009}$ |

in the intensive care unit (ICU) over a 48-hour period. Our objective is to model repeated measurements of 36 distinct attributes (we exclude weight from the original set of 37 attributes), including key physiological variables such as body temperature and heart rate. As auxiliary covariates, we incorporate $id$, $time$, $ICUtype$, $gender$, and $mortality$.

In our experiments, we utilize a subset of 3,997 patients provided in "set A" of the dataset. We randomly select 200 subjects for the validation and test sets, allocating 100 subjects to each. For these subjects, the first 10 time points are included in the training set to learn individual-specific temporal structure. The remaining time points are reserved for validation and testing. As shown in Table 3, DGBFGP achieves the best performance. While previous models perform close to the mean prediction, our approach achieves a significantly lower MSE value. These results demonstrate the effectiveness of our method in real-world forecasting tasks. Figure 13 illustrates the learned latent functions and their Sobol indices. The dominant $f_{ca}^{(1)}(id)$ accounts for heavy individual variation, while the others yield modest but significant capacity gains. Unsurprisingly, the $id$ term prevails by encoding each instance's defining traits.

*Table 4.* Conditional generation MSE on unseen character combinations of the SPRITES dataset.

| Method | Latent dimension | MSE |
|---|---|---|
| SGP-BAE (Tran et al., 2023) | 64 | $0.0145 \pm 0.0021$ |
| SVGP-VAE (Jazbec et al., 2021) | 64 | $0.0081 \pm 0.0005$ |
| LVAE (Ramchandran et al., 2021) | 64 | $0.0062 \pm 0.0004$ |
| DGBFGP (our work) | 64 | $\mathbf{0.0024 \pm 0.0003}$ |

*Table 5.* MSE comparison for evaluating semi-amortization and generalization across training setups.

| Training setup | Method | Rotated MNIST | Health MNIST | Physionet |
|---|---|---|---|---|
| Standard | SA-DGBFGP | $0.010 \pm 0.0004$ | $0.013 \pm 0.0005$ | $0.763 \pm 0.007$ |
| | DGBFGP* | $\mathbf{0.009 \pm 0.0001}$ | $\mathbf{0.009 \pm 0.0000}$ | $\mathbf{0.619 \pm 0.009}$ |
| Independent test subjects | SA-DGBFGP | $0.013 \pm 0.0004$ | $0.015 \pm 0.0006$ | $0.779 \pm 0.003$ |
| | DGBFGP* | $0.071 \pm 0.0002$ | $0.039 \pm 0.0003$ | $0.796 \pm 0.007$ |
| | DGBFGP | $\mathbf{0.009 \pm 0.0003}$ | $\mathbf{0.011 \pm 0.0005}$ | $\mathbf{0.658 \pm 0.026}$ |

## 5.4. SPRITES

The SPRITES dataset (Yingzhen & Mandt, 2018) comprises 1,296 unique characters, each defined by four categorical attributes: skin color, hairstyle, top clothing, and bottom clothing, with six possible options per attribute. Additionally, each character can perform actions from three categories, each executed in three directions over $T = 8$ time points. This results in a total of $N = 1,296 \times 9 \times 8$ images, each of size $D = 64 \times 64 \times 3$. The auxiliary covariates used in this experiment are $time, skin, hair, top, bottom, action$, and $direction$.

In this experiment, we aimed to demonstrate that our model effectively captures shared attributes defining each unique subject across a population without relying on explicit subject IDs. To evaluate this capability, we reserve 296 characters and all their action-time combinations for the test set, while another 296 characters are allocated to the validation set. The remaining characters are used for training.

Our method achieves noticeably lower MSE than previous approaches, as shown in Table 4. It also generates the most accurate SPRITES characters, as illustrated in Figure 4. These results highlight DGBFGP's ability to capture shared attributes and conditionally generate correlated character trajectories with unseen attribute combinations. Figure 15 shows example latent functions with normalized Sobol indices for each component. Unlike previous experiments, action and direction components dominate, highlighting that the latent representation encodes variability tied to those covariates. Hairstyle, clothing, and skin color contribute moderately, while time alone has minimal influence.

Table 10 shows that DGBFGP is the fastest model in all experiments among relevant approaches. This demonstrates its improved computational efficiency in addition to more accurate conditional generation. The results were obtained by running models on a single NVIDIA Tesla V100 GPU with 32GB of memory.

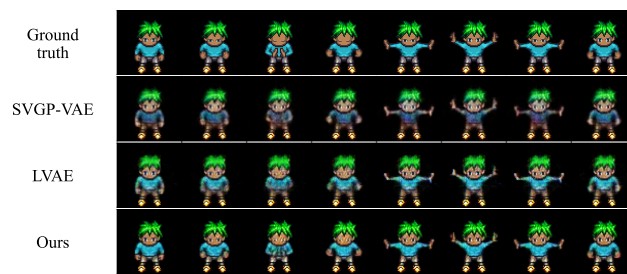

*Figure 4.* Conditionally generated character images from unseen attribute combinations of the SPRITES dataset. An additional example given in Appendix Figure 14.

## 5.5. Ablation of local latent variable amortization

In this experiment, we aim to determine whether learning a global encoder for instance-specific characteristics affects overall performance positively or negatively in the standard training setup, where at least some samples of the test subjects are included in the training set. Moreover, we assessed the generalization abilities of SA-DGBFGP by completely excluding the training samples of the test subjects from the training set and comparing its performance to both the non-finetuned DGBFGP model (denoted as DGBFGP*) and fine-tuned DGBFGP that has $L$ additional parameters for each new subject (Appendix B.1).

Table 5 displays the MSE values across various datasets. Under our standard training setup, SA-DGBFGP performs similarly to DGBFGP*, with only a slight performance decrease observed in the simulated datasets. However, this trend does not hold for the Physionet dataset, where semi-amortization exhibits poor performance. This discrepancy may be attributed to the amortization gap. When it comes to generalization abilities, DGBFGP* cannot perform as well as SA-DGBFGP as expected. However, our proposed fine-tuned DGBFGP outperforms the semi-amortized model.

## 6. Conclusions

In this work, we introduce a scalable basis-function approximation for GP-prior VAEs that eliminates the usual cubic-time bottleneck. By adopting a global parameterization, our method avoids explicit kernels, runs in linear time, eliminates the amortization gap, overcomes inducing point optimization constraints, supports standard mini-batch training, treats kernel hyperparameters probabilistically, and enhances latent space interpretability via Sobol indices. On various datasets, our approach yields markedly better predictive performance in conditional generation tasks than current state-of-the-art models. We believe this technique will prove invaluable for applications involving correlated samples where accuracy and efficiency are essential.

## Acknowledgements

We would like to acknowledge the computational resources provided by the Aalto Science-IT. We would like to thank the anonymous reviewers for their thoughtful comments that strengthened this work. This work was supported by the Research Council of Finland (decision number: 359135).

## Impact Statement

This paper introduces a scalable approach for analyzing high-dimensional time-series data, designed to improve computational efficiency, accuracy, and applicability for large-scale correlated datasets. We have not identified any potential negative societal impacts associated with this work.

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

## A. Other relevant background information

**Linear Models.** For simplicity and interpretability, one can assume that latent variables $z$ can be modeled as a linear combination of auxiliary covariates $x$ by a standard linear model:

$$z = a_1 x_1 + \cdots + a_Q x_Q + \epsilon = Ax + \epsilon,$$

where $a_q \in \mathbb{R}^L$, $A = [a_1, \ldots, a_Q] \in \mathbb{R}^{L \times Q}$ and $\epsilon$ is a noise that comes from a normal distribution with equal variance across $L$ dimensions.

It is possible to extend the capabilities of linear models and allow them to model nonlinear relationships by utilizing nonlinear basis functions. A basis function extension for latent variables using Fourier features was considered in (Ong et al., 2024).

## B. Model details

A covariate component $x^{(r)}$ can correspond to:

- A single continuous variable. In that case, the number of basis functions is $B^{(r)} = M^{(r)}$.

- Two variables (one continuous and one categorical with $C^{(r)}$ many possible values). In that case, the number of basis functions is $B^{(r)} = M^{(r)} C^{(r)}$.

- A single categorical variable. In that case, the number of basis functions is $B^{(r)} = C^{(r)}$, which in our work is exclusively used for ID covariate such that $C^{(r)} = P$.

The associated basis functions are represented as $\phi(x^{(r)}) \in \mathbb{R}^{B^{(r)}}$. The corresponding weight matrix $A^{(r)}$ is

$$A^{(r)} = \begin{bmatrix} a_{r1}^\top \\ \vdots \\ a_{rL}^\top \end{bmatrix} = \begin{bmatrix} a_{rlb} \end{bmatrix}_{L \times B^{(r)}},$$

where $l$ and $b$ represent the $l^{\text{th}}$ latent dimension and $b^{\text{th}}$ basis function, respectively. The collection of weights of all components can be represented as

$$A = \begin{bmatrix} A^{(1)} & \cdots & A^{(R)} \end{bmatrix} = \begin{bmatrix} a_{1l}^\top \ldots a_{Rl}^\top \end{bmatrix}_{L \times Q^+}.$$

Here, $Q^+ = \sum_{r=1}^R B^{(r)}$ indicates the size of the auxiliary covariates after applying basis functions.

### B.1. Modeling choice for IDs

In our experiments, we did not model interactions between IDs and a continuous variable due to scalability concerns, as the number of basis functions would grow to $M \times P$, significantly increasing computational complexity. Instead, we modeled IDs separately using a diagonal categorical kernel (Appendix Section D.2). Following the model presented in Section 4.3, the basis functions then correspond to one-hot encodings, and the linear model weights are sampled from the standard multivariate normal distribution. This approach simplifies the model while maintaining a manageable number of weights.

A trained model cannot be applied directly to a completely new instance with an unseen ID. However, retraining the entire model from scratch is unnecessary. Since covariates other than the ID are shared across all instances, we only need to fine-tune additional $L$ parameters to capture instance-specific characteristics for the new ID while keeping all other parameters fixed.

## C. Derivation of ELBO

Let $\boldsymbol{\Psi} = (A, \boldsymbol{\sigma}, \boldsymbol{\ell})$ be the generic notation for all unobserved random variables and $q(\boldsymbol{\Psi}) = q(A, \boldsymbol{\sigma}, \boldsymbol{\ell}) = q(A)q(\boldsymbol{\sigma})q(\boldsymbol{\ell})$. We adopted a mean-field approximation and employed the following factorizations for each variational distribution

$$q(A) = \prod_{r=1}^{R}\prod_{l=1}^{L} q(\boldsymbol{a}_{rl}) = \prod_{r=1}^{R}\prod_{l=1}^{L}\prod_{b=1}^{B^{(r)}} \mathcal{N}\left(a_{rlb} \mid \mu_{rlb}, \sigma_{rlb}^2\right) \tag{8}$$

$$q(\boldsymbol{\sigma}) = \prod_{r=1}^{R} \text{Lognormal}\left(\sigma_r \mid \mu_{r\sigma_q}, \sigma_{r\sigma_q}^2\right) \tag{9}$$

$$q(\boldsymbol{\ell}) = \prod_{r=1}^{R} \text{Lognormal}\left(\ell_r \mid \mu_{r\ell_q}, \sigma_{r\ell_q}^2\right). \tag{10}$$

The derivation of ELBO $\log p(Y \mid X) \geq \mathcal{L}(\boldsymbol{\Psi} \mid X, Y)$ is given as follows

$$\begin{aligned}
\mathcal{L}(\boldsymbol{\Psi} \mid X, Y) &= \int q(\boldsymbol{\Psi}) \log \frac{p_\varphi(Y \mid \boldsymbol{\Psi}, X)p(\boldsymbol{\Psi})}{q(\boldsymbol{\Psi})} d\boldsymbol{\Psi} \\
&= \mathbb{E}_{q(\boldsymbol{\Psi})}[\log p_\varphi(Y \mid \boldsymbol{\Psi}, X)] + \mathbb{E}_{q(\boldsymbol{\Psi})}[\log p(\boldsymbol{\Psi})] - \mathbb{E}_{q(\boldsymbol{\Psi})}[\log q(\boldsymbol{\Psi})] \\
&= \mathbb{E}_{q(A)}[\log p_\varphi(Y \mid A, X)] \\
&\quad + \mathbb{E}_{q(\boldsymbol{\sigma})q(\boldsymbol{\ell})q(A)}\left[\log \frac{p(A \mid \boldsymbol{\sigma}, \boldsymbol{\ell})}{q(A)}\right] + \mathbb{E}_{q(\boldsymbol{\sigma})}[\log p(\boldsymbol{\sigma})] + \mathbb{E}_{q(\boldsymbol{\ell})}[\log p(\boldsymbol{\ell})] \\
&\quad - \mathbb{E}_{q(\boldsymbol{\sigma})}[\log q(\boldsymbol{\sigma})] - \mathbb{E}_{q(\boldsymbol{\ell})}[\log q(\boldsymbol{\ell})] \\
&= \mathbb{E}_{q(A)}\left[\sum_{n=1}^{N} \log p_\varphi(\boldsymbol{y}_n \mid A, \boldsymbol{x}_n)\right] \\
&\quad - \mathbb{E}_{q(\boldsymbol{\sigma})q(\boldsymbol{\ell})}[\text{KL}(q(A) \mid\mid p(A \mid \boldsymbol{\sigma}, \boldsymbol{\ell}))] - \text{KL}(q(\boldsymbol{\sigma}) \mid\mid p(\boldsymbol{\sigma})) - \text{KL}(q(\boldsymbol{\ell}) \mid\mid p(\boldsymbol{\ell})),
\end{aligned}$$

where, in order to estimate $\mathbb{E}_{q(\boldsymbol{\sigma})q(\boldsymbol{\ell})}[\text{KL}(q(A) \mid\mid p(A \mid \boldsymbol{\sigma}, \boldsymbol{\ell}))]$, we sample $\boldsymbol{\sigma}$ and $\boldsymbol{\ell}$ from the corresponding distributions $q(\boldsymbol{\sigma})$ and $q(\boldsymbol{\ell})$ and denote these samples as $\hat{\boldsymbol{\sigma}}$ and $\hat{\boldsymbol{\ell}}$. These values are then substituted into spectral density computation, which we denote as $\hat{\boldsymbol{s}}$. Assuming a single Monte Carlo sample and using the mean-field assumptions from Equations (8)-(10) as well as closed-form expressions for KL divergencies between normal and log-normal distributions, the ELBO can be written as follows:

$$\begin{aligned}
\mathcal{L}(\boldsymbol{\Psi} \mid X, Y) &\approx \mathbb{E}_{q(A)}\left[\sum_{n=1}^{N} \log p_\varphi(\boldsymbol{y}_n \mid A, \boldsymbol{x}_n)\right] - \sum_{r=1}^{R}\sum_{l=1}^{L}\sum_{b=1}^{B^{(r)}} \text{KL}(q(\boldsymbol{a}_{rlb}) \mid\mid p(\boldsymbol{a}_{rlb}|\hat{\boldsymbol{\sigma}}_r, \hat{\boldsymbol{\ell}}_r)) \\
&\quad - \sum_{r=1}^{R} \text{KL}(q(\sigma_r) \mid\mid p(\sigma_r)) - \sum_{r=1}^{R} \text{KL}(q(\ell_r) \mid\mid p(\ell_r)) \\
&= \mathbb{E}_{q(A)}\left[\sum_{n=1}^{N} \log p_\varphi(\boldsymbol{y}_n \mid A, \boldsymbol{x}_n)\right] \\
&\quad - \sum_{r=1}^{R}\sum_{l=1}^{L}\sum_{b=1}^{B^{(r)}} \frac{\mu_{rlb}^2 + \sigma_{rlb}^2}{2\hat{s}_{rb}} - \frac{1}{2} - \log \frac{\sigma_{rlb}}{\sqrt{\hat{s}_{rb}}} \tag{11} \\
&\quad - \sum_{r=1}^{R} \frac{\sigma_{r\sigma_q}^2 + \mu_{r\sigma_q}^2 - 1}{2} - \log \sigma_{r\sigma_q} \quad - \sum_{r=1}^{R} \frac{\sigma_{r\ell_q}^2 + \mu_{r\ell_q}^2 - 1}{2} - \log \sigma_{r\ell_q}. \tag{12}
\end{aligned}$$

## D. Kernels

### D.1. Squared exponential kernel

The SE kernel, also known as the Gaussian kernel and the exponentiated quadratic kernel, is a stationary kernel used for continuous variables. It measures similarity between points based on their Euclidean distance, assuming the modeled function is smooth. It is formulated as follows:

$$k_{\text{se}}(x, x' \mid \sigma, \ell) = \sigma^2 \exp(-\frac{\|x - x'\|^2}{2\ell^2}),$$

where $\sigma^2$ is the variance determining the magnitude of the variations and $\ell$ is the length scale determining how quickly correlation decays with distance. The spectral density of the SE kernel is given by

$$s_{\text{se}}(\omega \mid \sigma, \ell) = \sigma^2 \ell \sqrt{2\pi} \exp(-\frac{\ell^2 \omega^2}{2}).$$

### D.2. Diagonal categorical kernel

The diagonal kernel is a simple kernel used for categorical variables. It measures similarity by checking whether two inputs belong to the same category. This kernel is formulated as:

$$k_{\text{ca}}(x, x') = \begin{cases} 1 & \text{if } x = x' \\ 0 & \text{otherwise.} \end{cases}$$

This kernel assigns a similarity of 1 to inputs from the same category and 0 otherwise. It is used when categories are independent and no shared structure is assumed between them.

### D.3. Zero-sum categorical kernel

The zero-sum kernel is also used for categorical variables. It ensures that the sum of similarities across all categories is zero. It is defined as:

$$k_{\text{zs}}(x, x') = \begin{cases} 1 & \text{if } x = x' \\ -\frac{1}{C-1} & \text{otherwise,} \end{cases}$$

where $C$ is the total number of categories. This kernel assigns a similarity of 1 for the same category and distributes the remaining similarity equally among all other categories as $-\frac{1}{C-1}$. It is suitable for modeling categorical variables where some correlation between different categories is expected while maintaining a sum-to-zero property.

### D.4. Other kernels

While our experiments were carried out using the standard kernels described above, it is straightforward to extend the proposed DGBFGP model to incorporate other kernels as well. The only requirements are that the covariance function for each continuous covariate is stationary, such that the eigenfunction approximation is valid and the spectral density exists, and that the covariance function for each categorical covariate is symmetric, such that the matrix eigendecomposition exists. If a dataset contains non-stationarities, it is still possible to apply the proposed DGBFGP model by using e.g. stationary kernels with input warpings (Snoek et al., 2014).

## E. Semi-amortized DGBFGP

When a new observation arises, we typically do not add additional categories for variables like biological sex. This rule does not apply to covariates that uniquely identify individuals, such as ID numbers, as each new individual necessarily brings a brand-new category value. In such a case, we need to re-train the whole model from scratch or, preferably, optimize additional $L$ parameters as explained in Section B.1. Another approach to tackle the generalization problem is learning a global function that predicts instance-specific latent variables using AVI while keeping global parameterizations for other covariates. The generative model remains exactly the same model as in DGBFGP. The only change is in the variational inference of the instance-specific part of parameters $A$. Without loss of generality, we assume that the $R^{\text{th}}$ component

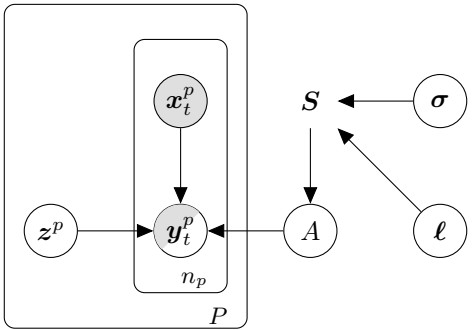

*Figure 5.* Graphical model of SA-DGBFGP.

in DGBFGP is the component responsible of modeling instance-specific characteristics. Accordingly, this component is removed from the global parameters and denoted as $Z = (\boldsymbol{z}^1, \ldots, \boldsymbol{z}^P)$ in SA-DGBFGP to be treated as a local, amortized, instance-specific parameter. With a slight misuse of notation, we use $A$ to denote $(A^{(1)}, \ldots, A^{(R-1)})$ below. SA-DGBFGP whose plate diagram given in Figure 5 is expressed as follows:

$$\sigma_r \sim \text{Lognormal}(0, 1)$$
$$\ell_r \sim \text{Lognormal}(0, 1)$$
$$\boldsymbol{S}_r(\sigma_r, \ell_r) = \text{diag}\big(\{s_{\text{se}}(\sqrt{\lambda_m} \mid \sigma_r, \ell_r\}_{m=1}^M\big)$$
$$A \mid \boldsymbol{\sigma}, \boldsymbol{\ell} \sim \prod_{r=1}^{R-1} \prod_{l=1}^{L} \mathcal{N}(\boldsymbol{a}_{rl} \mid \boldsymbol{0}, \boldsymbol{S}_r(\sigma_r, \ell_r))$$
$$Z \sim \prod_{p=1}^{P} \mathcal{N}(\boldsymbol{z}^p \mid \boldsymbol{0}, I_L)$$
$$Y \mid A, X \sim \prod_{n=1}^{N} p_\varphi(\boldsymbol{y}_n \mid A, \boldsymbol{z}_n, \boldsymbol{x}_n).,$$

We assume factorizing variational posterior, $q(A, Z, \boldsymbol{\sigma}, \boldsymbol{\ell} \mid Y_1) = q(A)q_\phi(Z \mid Y_1)q(\boldsymbol{\sigma})q(\boldsymbol{\ell})$, where $Y_1 = \{\boldsymbol{y}_1^p\}_{p=1}^P$.[1] We adopt the same mean-field assumptions given in Equations (8)-(10) for global parameters, and for local latent variables, we make use of the sample belonging to the first time point of an instance and assume the following factorization:

$$q_\phi(Z \mid Y_1) = \prod_{p=1}^{P} \mathcal{N}(\boldsymbol{z}^p \mid \boldsymbol{\mu}_\phi(\boldsymbol{y}_1^p), \boldsymbol{\sigma}_\phi^2(\boldsymbol{y}_1^p)I_L).$$

Therefore, the ELBO objective of SA-DGBFGP becomes:

$$\log p(Y \mid X) \geq \sum_{p=1}^{P} \left( \sum_{t=1}^{n_p} \mathbb{E}_{q(A)q(\boldsymbol{z}^p \mid \boldsymbol{y}_1^p)} \left[ \log p_\varphi(\boldsymbol{y}_t^p \mid A, \boldsymbol{z}^p, \boldsymbol{x}_t^p) \right] \right) - \text{KL}(q_\phi(\boldsymbol{z}^p \mid \boldsymbol{y}_1^p) \mid\mid p(\boldsymbol{z}^p))$$
$$- \mathbb{E}_{q(\boldsymbol{\sigma})q(\boldsymbol{\ell})}[\text{KL}(q(A) \mid\mid p(A \mid \boldsymbol{\sigma}, \boldsymbol{\ell}))]$$
$$- \text{KL}(q(\boldsymbol{\sigma}) \mid\mid p(\boldsymbol{\sigma}))$$
$$- \text{KL}(q(\boldsymbol{\ell}) \mid\mid p(\boldsymbol{\ell})).$$

We note that using the standard normal distribution as the prior for local latent variables is analogous to predicting global linear parameters that affect a specific instance based on its first sample. This is because, when using the diagonal categorical

---

[1] We also experimented with utilizing all observed time points, but the results were not better than those obtained using only the first time point.

kernel defined in Section D.2, both the $\boldsymbol{D}$ and $\boldsymbol{\Theta}$ matrices are identity matrices of size $C$, $I_C$, which activate only the $c^{\text{th}}$ entry that is related to the category $c$ in global parameter vector, $\boldsymbol{a}$. Hence, we can plug inferred latent variables into our additive structure without any concern.

## F. Experimental details

Since we specified a general Lognormal$(0, 1)$ prior for the lengthscale parameters, we standardized all continuous auxiliary covariates to have zero mean and unit variance exclusively for training DGBFGP. In contrast, other GP prior VAEs do not impose a prior on kernel hyperparameters, as these are optimized jointly with the neural network parameters (Jazbec et al., 2021; Ramchandran et al., 2021). Regardless of the approach, samples are scaled to the range $[0, 1]$ in the Rotated MNIST, Health MNIST, and SPRITES experiments, while they are standardized in the Physionet experiment.

In all experiments, we used a fixed number of basis functions, $M = 10$, to approximate continuous kernels. Additionally, for all experiments, the boundary condition $\Omega = [-J, J]$ was defined such that $J$ is approximately $c$ times half the range of the corresponding continuous values, following the recommendations in (Riutort-Mayol et al., 2023; Timonen & Lähdesmäki, 2024). The approximation with a finite number of basis functions is expected to be effective as long as $J$ is not too close to the absolute maximum of the continuous covariate (Riutort-Mayol et al., 2023). Based on this criterion, we set $c = 1.5$.

We implemented our model using PyTorch (Paszke et al., 2019). Across all experiments and methods, we employed the Adam optimizer (Kingma, 2014) with a learning rate of 0.001 and a batch size of 256. To ensure stable convergence, the learning rate was dynamically reduced whenever the validation loss did not improve for a specified number of epochs. Specifically, the learning rate was scaled down by a factor of 0.5 if no improvement was observed for 10 consecutive epochs. Additionally, early stopping was employed to prevent overfitting, and the final model for each experiment was selected based on the best performance on the respective validation set.

In the remainder of this section, we provide detailed specifications for each experiment. We denote the additive components utilizing different covariance functions as follows: $f_{\text{se}}(\cdot)$ for the SE kernel, $f_{\text{se}\times\text{ca}}(\cdot)$ for the interaction of SE and CA kernels, and $f_{\text{ca}}(\cdot)$ for the CA kernel which is applied exclusively to the $id$ covariate, when applicable.

### F.1. Rotated MNIST

For the validation and test sets, we randomly selected 80 distinct instances and randomly sampled four consecutive angles for every instance. The remaining samples were allocated to the training set. As a result, the training set comprises $N_{\text{train}} = 5,760$ samples, while both the validation and test sets contain $N_{\text{val}} = N_{\text{test}} = 320$ samples each.

For the Rotated MNIST experiment, we modeled the latent space using the following additive kernel structure:

$$f = f_{\text{ca}}^{(1)}(\text{id}) + f_{\text{se}}^{(2)}(\text{angle}).$$

In contrast, LVAE (Ramchandran et al., 2021) employed a more complex setup that incorporates an additional component that models interactions between $id$ and the continuous $angle$ covariate, $f_{\text{se}\times\text{ca}}(\text{angle} \times \text{id})$, as this configuration was reported to be the most effective in the original work (Ramchandran et al., 2021).

The maximum number of epochs for this experiment was set to $2,000$.

### F.2. Health MNIST

The kernel structure for the Health MNIST experiment was defined as:

$$f = f_{\text{ca}}^{(1)}(\text{id}) + f_{\text{se}}^{(2)}(\text{time}) + f_{\text{se}\times\text{ca}}^{(3)}(\text{time} \times \text{gender}) + f_{\text{se}\times\text{ca}}^{(4)}(\text{diseaseTime} \times \text{disease}).$$

Similarly, as in the Rotated MNIST experiment, LVAE used an additional interaction component $f_{\text{se}\times\text{ca}}(\text{time} \times \text{id})$.

The maximum number of epochs for this experiment was also set to $2,000$.

### F.3. Physionet

For the Physionet experiment, our model utilized the following kernel structure:

$$
\begin{aligned}
f =& f_{\mathrm{ca}}^{(1)}(\mathrm{id}) + f_{\mathrm{se}}^{(2)}(\mathrm{time}) + f_{\mathrm{se\times ca}}^{(3)}(\mathrm{time} \times \mathrm{ICUtype}) \\
& + f_{\mathrm{se\times ca}}^{(4)}(\mathrm{time} \times \mathrm{gender}) + f_{\mathrm{se\times ca}}^{(5)}(\mathrm{time} \times \mathrm{mortality}),
\end{aligned}
$$

whereas LVAE included an additional interaction term, $f_{\mathrm{se\times ca}}(\mathrm{time} \times \mathrm{id})$.

The maximum number of epochs for this experiment was set to $1,000$.

### F.4. SPRITES

For the SPRITES experiment, both our model and LVAE used the same kernel structure:

$$
\begin{aligned}
f =& f_{\mathrm{se}}^{(1)}(\mathrm{time}) + f_{\mathrm{se\times ca}}^{(2)}(\mathrm{time} \times \mathrm{skin}) + f_{\mathrm{se\times ca}}^{(3)}(\mathrm{time} \times \mathrm{bottom}) \\
& + f_{\mathrm{se\times ca}}^{(4)}(\mathrm{time} \times \mathrm{top}) + f_{\mathrm{se\times ca}}^{(5)}(\mathrm{time} \times \mathrm{hair}) \\
& + f_{\mathrm{se\times ca}}^{(6)}(\mathrm{time} \times \mathrm{action}) + f_{\mathrm{se\times ca}}^{(7)}(\mathrm{time} \times \mathrm{direction}).
\end{aligned}
$$

The maximum number of epochs for this experiment was set to 100.

## G. Neural network architectures

We present neural network architectures employed across different experiments. Table 6 details the convolutional neural network (CNN) architecture used for the Rotated MNIST experiment; it is similar to the one in (Casale et al., 2018), with minimal modifications. Likewise, Table 7 outlines the CNN architecture used for Health MNIST, closely following (Ramchandran et al., 2021). Table 8 describes the multi-layer perceptron (MLP) architecture employed for the Physionet experiment, and Table 9 shows the CNN-based architecture from (Jazbec et al., 2021) with a minor alteration in the final activation layer. In CNN-based architectures, we use a sigmoid activation function in the last layer of the generative network, which is better suited to our pre-processed data. By contrast, for the MLP-based architecture, we omit an activation function in the final layer because the processed data are not bounded.

*Table 6.* Neural network architecture used in Rotated MNIST experiment

|  | Hyperparameter | Value |
|---|---|---|
| Inference network | Dimensionality of input | $28 \times 28$ |
|  | Number of convolution layers | 3 |
|  | Number of filters per convolution layer | 72 |
|  | Kernel size | $3 \times 3$ |
|  | Stride | 2 |
|  | Number of feedforward layers | 1 |
|  | Width of feedforward layers | 128 |
|  | Dimensionality of latent space | $L$ |
|  | Activation function of layers | ELU |
| Generative network | Dimensionality of input | $L$ |
|  | Number of transposed convolution layers | 3 |
|  | Number of filters per transposed convolution layer | 72 |
|  | Kernel size | 3 |
|  | Stride | 2 |
|  | Number of feedforward layers | 1 |
|  | Width of feedforward layers | 128 |
|  | Activation function of layers | ELU, Sigmoid |

*Table 7.* Neural network architecture used in Health MNIST experiment

| | Hyperparameter | Value |
|---|---|---|
| Inference network | Dimensionality of input | $36 \times 36$ |
| | Number of convolution layers | 2 |
| | Number of filters per convolution layer | 16, 32 |
| | Kernel size | $3 \times 3$ |
| | Stride | 1 |
| | Pooling | Max pooling |
| | Pooling kernel size | $2 \times 2$ |
| | Pooling stride | 2 |
| | Number of feedforward layers | 2 |
| | Width of feedforward layers | 300, 30 |
| | Dimensionality of latent space | $L$ |
| | Activation function of layers | RELU |
| Generative network | Dimensionality of input | $L$ |
| | Number of transposed convolution layers | 2 |
| | Number of filters per transposed convolution layer | 16 |
| | Kernel size | $4 \times 4$ |
| | Stride | 2 |
| | Number of feedforward layers | 2 |
| | Width of feedforward layers | 30, 300 |
| | Activation function of layers | RELU, Sigmoid |

*Table 8.* Neural network architecture used in Physionet experiment

| | Hyperparameter | Value |
|---|---|---|
| Inference network | Dimensionality of input | 36 |
| | Number of feedforward layers | 2 |
| | Width of feedforward layers | 300, 30 |
| | Dimensionality of latent space | $L$ |
| | Activation function of layers | RELU |
| Generative network | Dimensionality of input | $L$ |
| | Number of feedforward layers | 2 |
| | Width of feedforward layers | 30, 300 |
| | Activation function of layers | RELU |

*Table 9.* Neural network architecture used in SPRITES experiment

| | Hyperparameter | Value |
|---|---|---|
| Inference network | Dimensionality of input | $3 \times 64 \times 64$ |
| | Number of convolution layers | 6 |
| | Number of filters per convolution layer | 16 |
| | Kernel size | $3 \times 3$ |
| | Stride | 1,2 |
| | Dimensionality of latent space | $L$ |
| | Activation function of layers | ELU |
| Generative network | Dimensionality of input | $L$ |
| | Number of convolution layers | 6 |
| | Number of filters per convolution layer | 16 |
| | Kernel size | $3 \times 3$ |
| | Stride | 1 |
| | Activation function of layers | ELU, Sigmoid |

# H. Additional results and comparisons

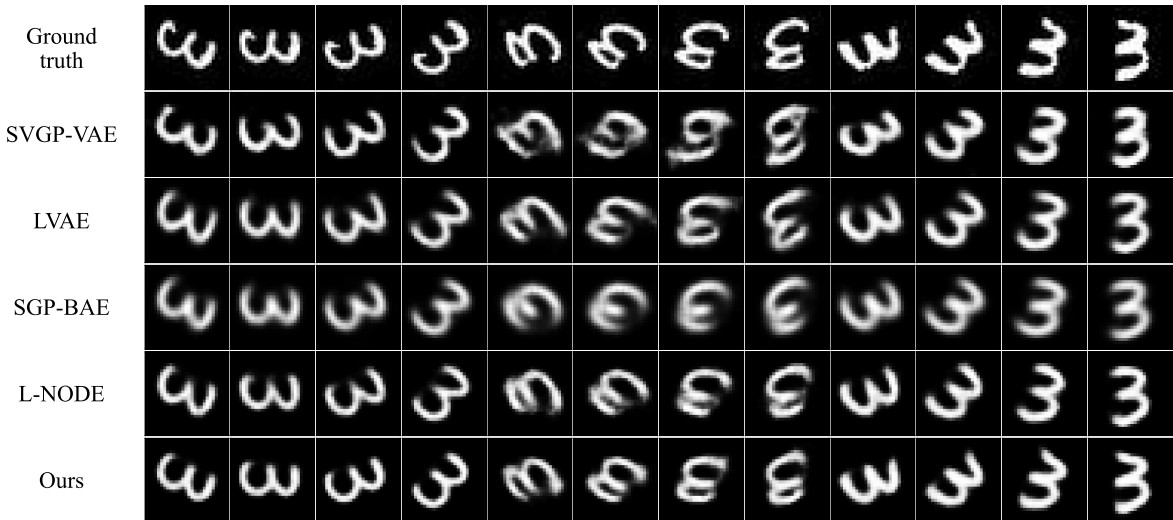

*Figure 6.* Conditionally generated images of three different test instances at unseen rotation angles in Rotated MNIST experiment.

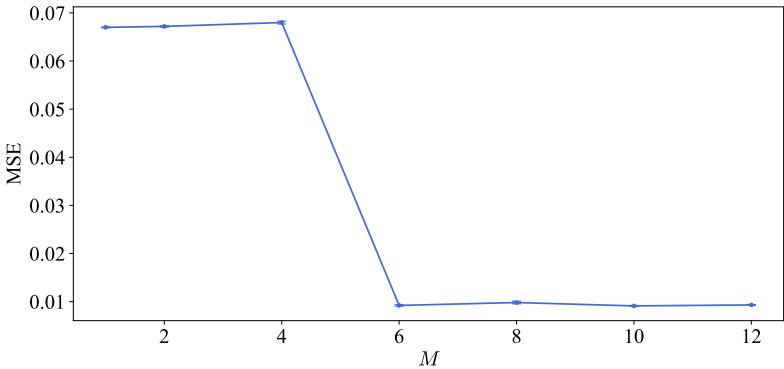

*Figure 7.* Test MSE as a function of $M$ in the Rotated MNIST experiment.

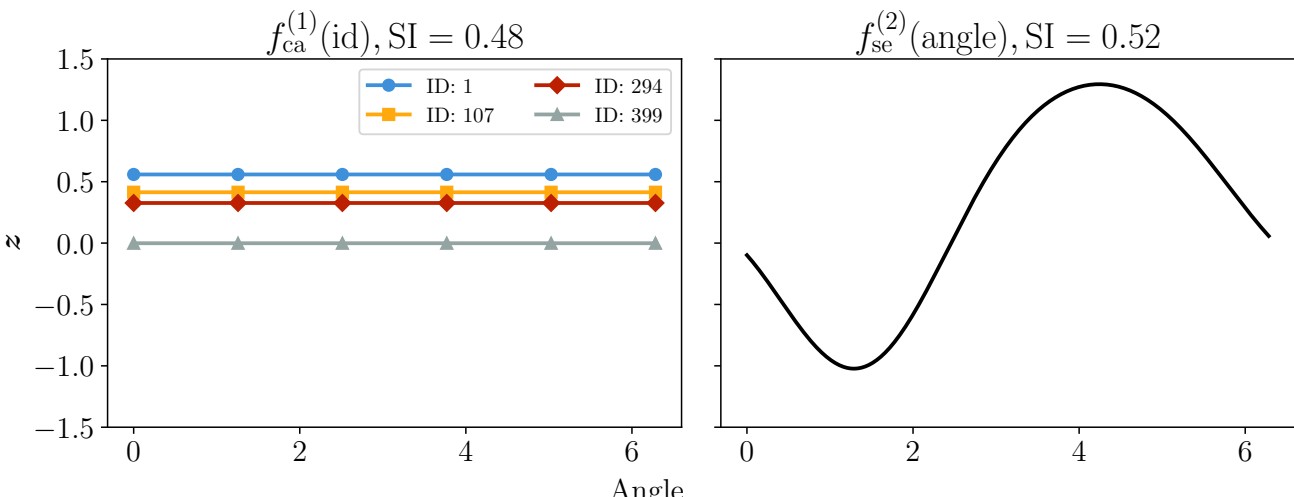

*Figure 8.* Example learned latent functions in the Rotated MNIST experiment along with normalized Sobol indices averaged over $L = 16$ dimensions.

*Table 10.* Run time comparison of GP prior VAEs per epoch in terms of seconds.

| Method | Rotated MNIST | Health MNIST | Physionet | SPRITES |
|---|---|---|---|---|
| LVAE (Ramchandran et al., 2021) | $7.7 \pm 0.1$ | $34.9 \pm 0.4$ | $85.2 \pm 0.9$ | $88.3 \pm 0.5$ |
| SVGP-VAE (Jazbec et al., 2021) | $1.8 \pm 0.1$ | $13.6 \pm 0.5$ | $80.7 \pm 0.5$ | $35.9 \pm 0.6$ |
| DGBFGP (our work) | $0.5 \pm 0.0$ | $12.5 \pm 0.2$ | $13.1 \pm 0.1$ | $10.2 \pm 0.4$ |

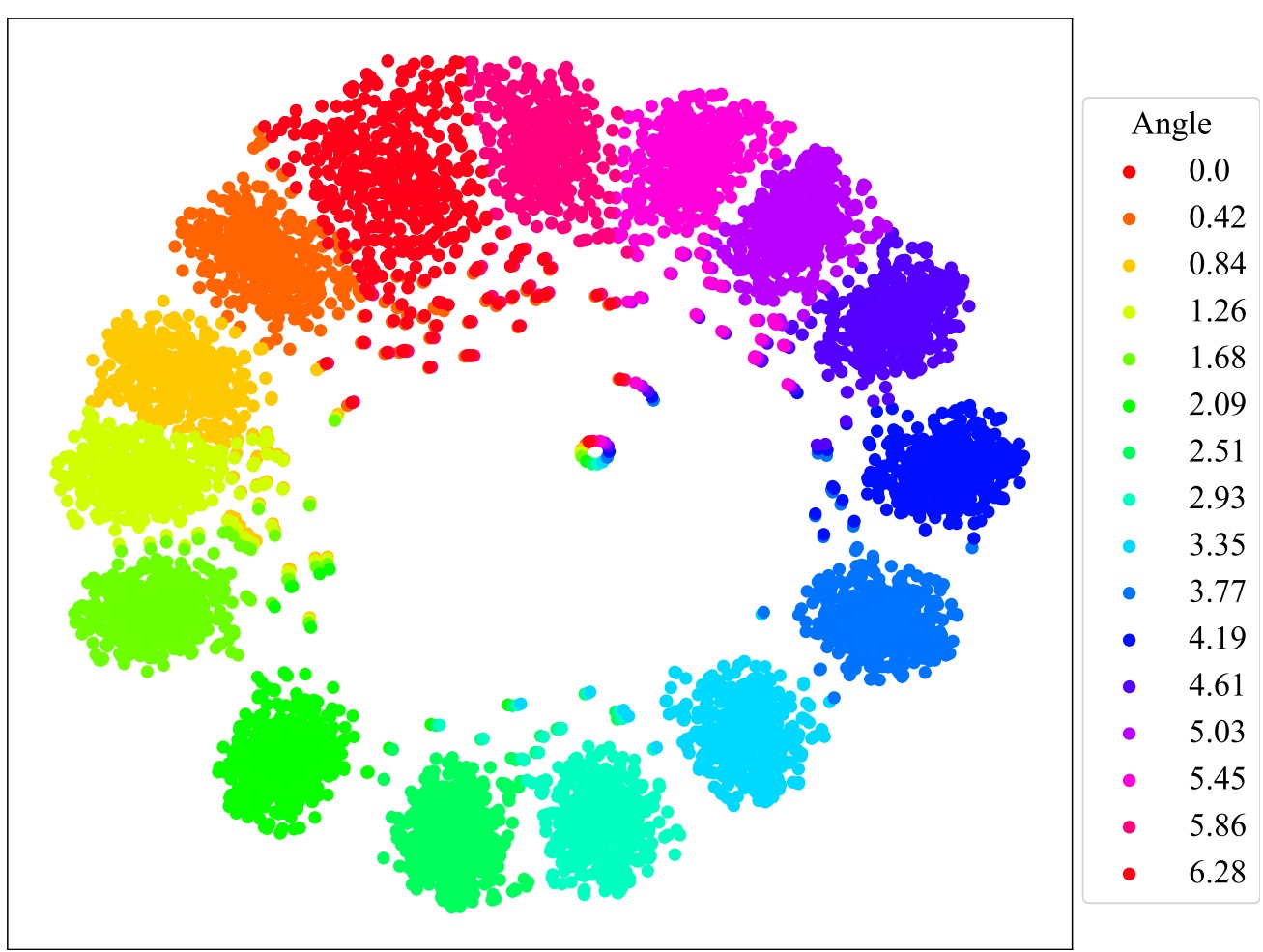

*Figure 9.* t-SNE embedding of the latent variables of the Rotated MNIST data.

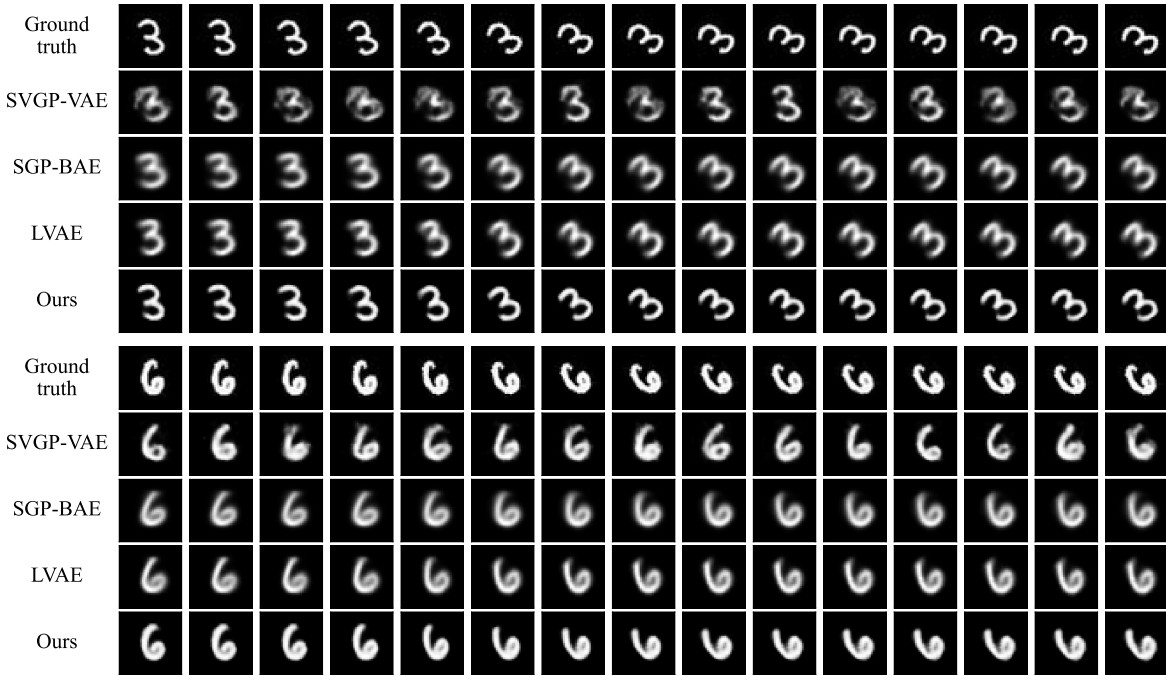

*Figure 10.* Conditionally generated images of two different test instances for future prediction on the Health MNIST dataset.

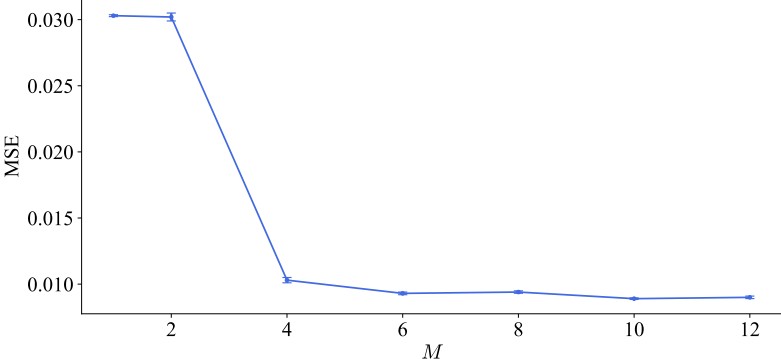

*Figure 11.* Test MSE as a function of $M$ in the Health MNIST experiment.

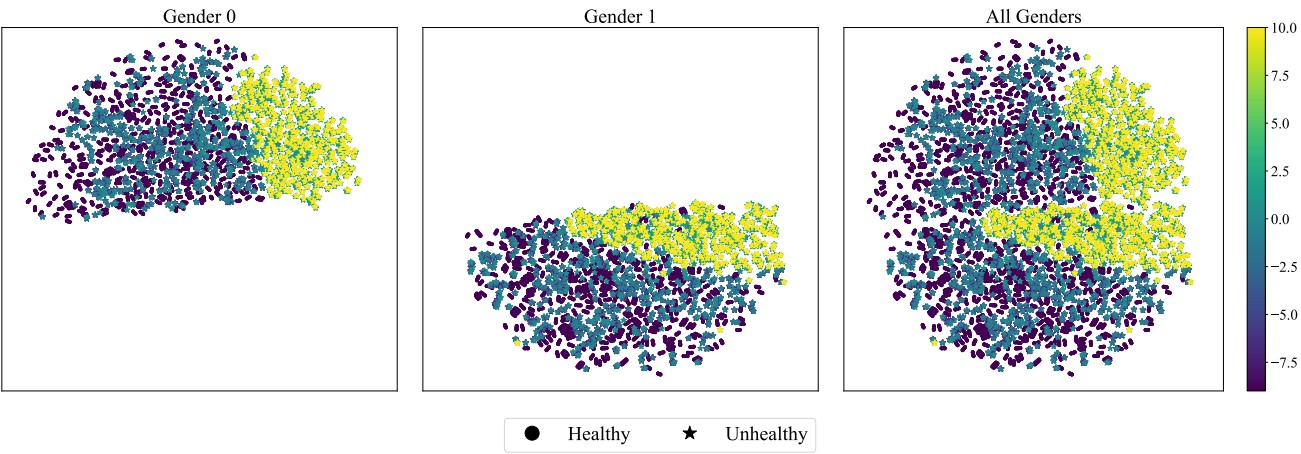

*Figure 12.* t-SNE embedding of the latent variables of the Health MNIST data, with the color bar representing the $diseaseTime$ covariate. This covariate is applicable exclusively to unhealthy subjects.

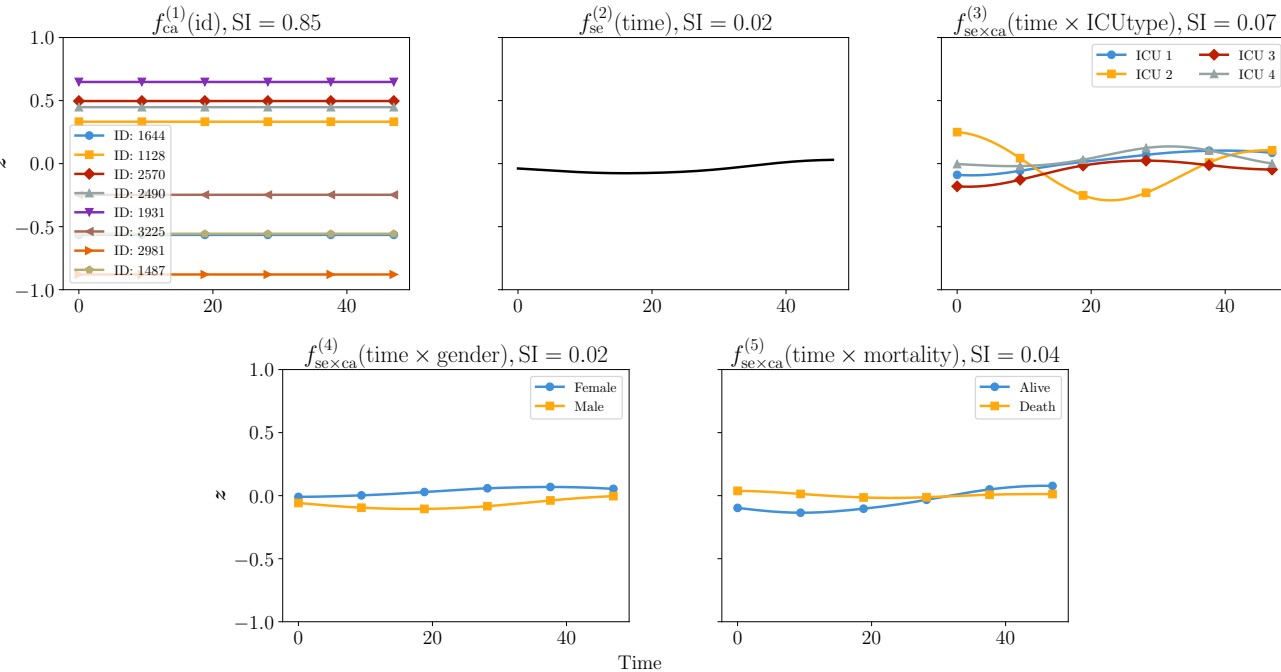

*Figure 13.* Example learned latent functions in the Physionet experiment along with normalized Sobol indices averaged over $L = 32$ dimensions.

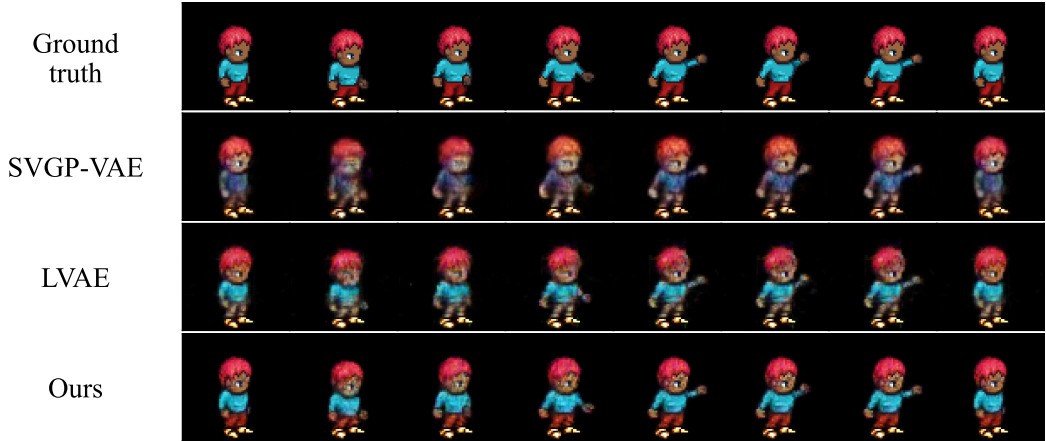

*Figure 14.* Additional example of generated character images from unseen combinations of the SPRITES dataset.

*Figure 15.* Example learned latent functions in the SPRITES experiment along with normalized Sobol indices averaged over $L = 64$ dimensions.

