# OpenReview forum: "Bayesian Basis Function Approximation for Scalable Gaussian Process Priors in Deep Generative Models"
_ICML.cc/2025/Conference — ICML 2025 poster_

### Official Review · Reviewer_tjXK · 2025-03-07

**Overall Recommendation:** 4

**Summary:**

This paper addresses the computational challenges of using Gaussian process (GP) priors in Variational Autoencoders (VAEs) for high-dimensional time series analysis. While GP-based VAEs effectively capture temporal dependencies, their cubic time complexity limits scalability. To overcome this limitation, the authors propose a scalable basis function approximation for additive GP priors in VAEs, reducing computational complexity to linear time. The authors claim that their approach is not only computationally efficient but also capable of capturing complex correlations within and across subjects. Empirical evaluations on synthetic and real-world datasets demonstrate that the proposed method enhances computational efficiency while significantly improving predictive performance compared to existing approaches.

**Claims And Evidence:**

The main claim of the paper on its proposed method include

1) Model Expressiveness: The proposed model, DGBFGP, is claimed to be as expressive as or more expressive than other GP-based VAEs in capturing complex correlations within and across subjects.

The authors support this claim by evaluating the model on both temporal interpolation and long-term forecasting tasks using standard benchmarks, including Rotated MNIST, Health MNIST, PhysioNet, and SPRITES. The results demonstrate a significant performance improvement over competing models. However, some relevant models are missing from the benchmark comparisons. Including these baselines would provide a more comprehensive and rigorous evaluation of DGBFGP’s expressiveness.

2) Computational efficiency: The paper evaluates computational efficiency by comparing worst-case complexity (Big-O notation), a standard metric for algorithmic runtime. The authors show that DGBFGP achieves a computational complexity of: $O(N\sum_{p}B^{r})$ which improves upon previous approaches such as Ramchandran et al. (2021) and Jazbec et al. (2021), which have a complexity of $O(NM^2+M^3)$ where M is the number of inducing points.

Additionally, empirical runtime comparisons are conducted across four datasets (Rotated MNIST, Health MNIST, PhysioNet, and SPRITES), demonstrating that DGBFGP runs significantly faster than LVAE. However, the evaluation is limited to LVAE, and it would be more comprehensive to include runtime comparisons with additional baselines for a more complete assessment of computational efficiency.

**Essential References Not Discussed:**

The paper provides a comprehensive overview of prior work on Gaussian Process (GP) priors in Variational Autoencoders (VAEs) and related applications. However, it does not cite the foundational work on Variational Inference (VI) by Hoffman et al. (2013), despite heavily relying on concepts from Stochastic Variational Inference (SVI) for scalable inference.

Hoffman, M. D., Blei, D. M., Wang, C., & Paisley, J. (2013). Stochastic Variational Inference. Journal of Machine Learning Research, 14(1), 1303–1347.

**Experimental Designs Or Analyses:**

The experimental design is generally appropriate. Authors evaluate both model expressiveness and computational efficiency across multiple datasets. The performance comparisons demonstrate significant improvements, but the absence of certain baseline models raises concerns about the completeness of the evaluation. Additionally, while the runtime analysis supports the claim of improved efficiency, it is only compared against LVAE, limiting broader conclusions. Including additional baselines for both performance and runtime would strengthen the validity of the results.

**Methods And Evaluation Criteria:**

The proposed methods and evaluation criteria are generally appropriate, but there are areas where the evaluation could be more comprehensive.

The authors evaluate model expressiveness by benchmarking DGBFGP on standard datasets (Rotated MNIST, Health MNIST, PhysioNet, and SPRITES), covering both temporal interpolation and long-term forecasting tasks. While the results show a significant performance improvement, the absence of certain baseline models makes it difficult to fully assess the model’s expressiveness. Including additional comparisons would strengthen the claims.

For computational efficiency, the paper provides both theoretical analysis (Big-O complexity) and empirical runtime comparisons. The authors demonstrate that DGBFGP reduces computational complexity relative to prior GP-based VAEs and empirically show that it runs significantly faster than LVAE. However, the runtime comparison is only performed against LVAE, and a more thorough evaluation against multiple baselines would provide a clearer understanding of the model’s efficiency across different settings.

**Other Comments Or Suggestions:**

I have no additional comments or suggestions beyond those already discussed in the other sections.

**Other Strengths And Weaknesses:**

The paper is well-written with a clear goal and a strong focus on improving GP-based VAEs for time-series forecasting. It effectively highlights its advancements over existing models but could benefit from stronger motivation for its approach.
The paper has a highly specific application: time-series forecasting using VAEs with Gaussian Process priors. It would be helpful to contextualize its relevance. Specifically, how does it compare to non-Transformer-based alternatives, and what are its broader implications and use cases? Addressing these questions would strengthen the paper’s impact and clarify its significance in the broader time-series modeling landscape.

**Questions For Authors:**

Expressiveness and the Role of $R$
1) The expressiveness of the model likely depends on $R$, the number of latent additive dimensions. How does the model's performance vary as a function of $R$?
2) I noticed that the latent dimensionality in experiments ranges from 16 to 64. Is there a specific reasoning behind this choice? Was there a trade-off considered between expressiveness, computational efficiency, and overfitting?
3) I may be misunderstanding aspects of the paper, but given that DGBFGP approximates the additive Gaussian Process kernels used in LVAE, what specifically allows it to achieve significantly better performance?
4) Is the improvement primarily due to a better approximation method, improved inference procedure, or other architectural differences?

**Relation To Broader Scientific Literature:**

The paper extends Variational Autoencoders (VAEs) with Gaussian Process (GP) priors for multivariate time-series modeling by introducing a Bayesian basis function approach to approximate mixed-domain additive Gaussian Processes. The authors provide a comprehensive review of existing GP-based VAEs for time-series applications (Sohn et al., 2015; Cao et al., 2018; Iakovlev et al., 2023; Casale et al., 2018; Fortuin et al., 2020; Jazbec et al., 2021; Ramchandran et al., 2021), highlighting the key challenges associated with each method. They then introduce their proposed model, DGBFGP, which addresses these limitations by improving scalability and efficiency while maintaining the expressiveness of GP-based VAEs.

**Theoretical Claims:**

No original theoretical claims are made in the paper.

---

> ### Author Rebuttal · Authors · 2025-03-31
>
> We thank the reviewer for their comments.
>
> **Missing relevant models.** Our experiments include previous GP-VAE methods, as our main goal is to improve these models. We also evaluate an RNN-based method (BRITS) and a latent neural ODE (L-NODE). We also compare against SGP-BAE suggested by reviewer VgcQ. We're happy to run additional comparisons if the reviewer specifies which method they refer to.
>
> **Lacking runtime comparisons.** Initially, we focused on LVAE due to its similarity to our method since only LVAE and DGBFGP can handle an arbitrary number of covariates. However, we agree that adding more baselines gives a bigger picture of DGBFGP's efficiency. We now include SVGP-VAE in our runtime comparison.
>
> |Method|R. MNIST|H. MNIST|Physionet|SPRITES|
> |-|-|-|-|-|
> LVAE|$7.7\pm0.1$| $34.9\pm0.4$|$85.2\pm0.9$|$88.3\pm0.5$|
> SVGP-VAE|$1.8\pm0.1$|$13.6\pm0.5$|$80.7\pm0.5$|$35.9\pm0.6$
> DGBFGP|$0.5\pm0.0$|$12.5\pm0.2$|$13.1\pm0.1$|$10.2\pm0.4$
>
> **VI reference is missing.** We agree. We will update the manuscript with an appropriate citation for SVI, ensuring our methodology is properly contextualized within the broader VI literature.
>
> **Stronger motivation.** Agreed. In the final version, we will expand our discussion to highlight existing models' limitations and how our model improves them. See our response "Contributions" for reviewer VgcQ for revised motivation detailing these limitations and our contributions.
>
> **Compare to non-transformer-based alternatives.**
> High-dimensional multivariate time series face challenges from complex correlations, time-varying covariates, and missing values. Standard VAEs with an iid normal prior often struggle with these aspects. By incorporating a GP prior into a VAE, our model: 1) Provides reliable uncertainty estimates without extra mechanisms (unlike state-space models or RNNs). 2) Uses tunable kernel functions for transparency, outperforming "black-box" approaches. 3) "filters out" noise by learning data generating process to reveal the underlying structure.
>
> Moreover, while ODE-based methods struggle with varying dynamics (e.g., between healthy and unhealthy subjects), our GP-based model remains robust. Experiments comparing it to BRITS and L-NODE underscore these advantages, making it especially beneficial for healthcare.
>
> **Answer for Q1.** We agree: our method's performance depends on R, but we view R and its latent components as a modeling choice rather than a hyperparameter. The expressiveness of DGBFGP stems from its additive components that model each covariate. For instance, the Health MNIST experiment in Appendix E.2 uses the following additive model:
>
> $$
> f_{\mathrm{ca}}^{(1)}(\mathrm{id})+f_{\mathrm{se}}^{(2)}(\mathrm{time})+f_{\mathrm{se \times ca}}^{(3)}(\mathrm{time \times gender})+f_{\mathrm{se \times ca}}^{(4)}(\mathrm{diseaseTime \times disease})
> $$
>
> Ablation results for the Health MNIST by removing some components from the original implementation:
>
> |Additive model|R|MSE|
> |-|-|-|
> |$f^{(1)}+f^{(2)}+f^{(3)}+f^{(4)}$|4|0.009|
> |$f^{(1)}+f^{(2)}+f^{(4)}$|3|0.010|
> |$f^{(1)}+f^{(2)}+f^{(3)}$|3|0.025|
> |$f^{(1)}+f^{(2)}$|2|0.025|
> |$f^{(1)}$|1|0.031|
> |$f^{(2)}$|1|0.036|
>
> Removing the 3rd component has little effect since the gender signal is indirectly captured by the digit type. In contrast, 4th component is crucial, as it captures the evolving disease signal; without it, the model cannot discern an instance's health trajectory. Similarly, removing time-related components weakens temporal correlation capture, and eliminating the id term forces all instances to share latent variables, thereby losing individual nuances. This analysis shows that while the numerical value of R doesn't directly dictate expressiveness, the design of each additive component is essential. We will include this analysis into the revised Appendix.
>
> **Answer for Q2.** Since the main focus of this paper is to model latent space using GP priors, for a fair comparison, we followed the architectures and latent dimensionaly choices provided in the previous GP prior VAE works as we denoted in Appendix F.
>
> **Answer for Q3 and Q4.** In models like LVAE and SVGP-VAE, inducing point placement is crucial because they summarize the latent function over the input space. If they don't cover all covariates, key variations can be missed, leading to suboptimal performance, especially with discrete covariates, whose locations can't be optimized with gradients.
>
> For fair comparisons, we used identical architectures across models, as our goal is to present an alternative GP prior approximation for latent space modeling. The Hilbert space approximation offers global parameterization, eliminating the need for a shared inference network and the associated amortization gap. As [1] shows that the amortization gap is unavoidable in latent variable models, including GPs, our method avoids the suboptimality of a shared encoder network.
>
> [1] Amortized Variational Inference: When and Why?
> https://arxiv.org/abs/2307.11018

---

> > ### Comment · Reviewer_tjXK · 2025-04-09
> >
> > Thank you for the thoughtful and thorough rebuttal. These updates address my main concerns, and I now lean toward acceptance.

---

### Official Review · Reviewer_L1D7 · 2025-03-12

**Overall Recommendation:** 2

**Summary:**

The authors present an approach for performing efficient variational inference in an additive GP prior VAE (additive GP prior with an MLP parameterised likelihood). The approach uses standard low-rank Hilbert space approximations for the GP kernels, approximating the additive GP prior as an additive linear model. The experimental results demonstrate that the proposed model and approximation outperform a number of GP-VAE baselines on several experiments.

**Claims And Evidence:**

The main claims are supported with evidence.

**Essential References Not Discussed:**

N/A.

**Experimental Designs Or Analyses:**

The experimental design follow that of previous papers.

**Methods And Evaluation Criteria:**

Yes. The method is evaluated on MNIST-derived, SPRITES, and Physionet based experiments, which it seems are standard for papers in this area.

**Other Comments Or Suggestions:**

N/A.

**Other Strengths And Weaknesses:**

## Strengths

* The paper is clearly written and easy to follow.
* The model and approximation method outperform the baselines across the experiments.

## Weaknesses

* Whilst the method is sensible and correct, I can't help but feel the paper is lacking novelty---the methods apply existing GP approximation techniques to an existing model.
* Although the global variational parameterisation is hailed as a strength of the method; because the variational approximation is not amortised it is unable to generalise to new distinct instances in e.g. the MNIST experiments.
* I presume that it is possible to use a sparse GP approximation for the additive GP prior VAE model? If so, it would be useful if a comparison was provided.

**Questions For Authors:**

N/A.

**Relation To Broader Scientific Literature:**

N/A.

**Theoretical Claims:**

No,

---

> ### Author Rebuttal · Authors · 2025-03-31
>
> **Lacking novelty.** We kindly ask the reviewer to find what we contribute to the literature by referring to our response "Contributions" for the reviewer VgcQ.
>
> **Generalization for new instances.** Thank you for giving us the opportunity to clarify this important aspect of our model. We agree with the reviewer in that it is absolutely essential that predictions for new distinct test instances are adapted to each instance. However, we respectfully disagree with the reviewer's claim that our method cannot do that. As detailed in Appendix B.1, our approach generalizes to new distinct test instances by variationally optimizing $L$ instance-specific id parameters using only the data from the test instance from the test instance while keeping all other model parameters fixed at test time, where $L$ is the latent space dimensionality. Given that we need to optimize only a small number of instance-specific variational parameters, it efficiently learns instance-specific characteristics without significant computational overhead. Moreover, our approach to generalize to new distinct test instances is also theoretically more accurate than the amortization-based approach as we can avoid the amortization gap when generalizing to new test instances.
>
> To clarify this further, we provide additional evidence that our proposed method indeed can generalize to new distinct test instances even better than methods that use amortized VI. In our experiments, the model variant where instance-specific parameters are optimized at test time corresponds to the proposed method and is denoted as DGBFGP, and that is compared to the amortized model variant as well as to a baseline model that does not have any mechanism to generalize to new test instances (denoted by DGBFGP$^*$).
> Our proposed model outperforms both the baseline and an amortized variant, as shown below:
>
> |Method|Rotated MNIST|Health MNIST|
> |-|-|-|
> |DGBFGP$^*$|$0.071 \pm 0.0002$|$0.039 \pm 0.0003$|
> Amortized DGBFGP | $0.013 \pm 0.0006$ | $0.015 \pm 0.0005$
> DGBFGP (proposed model) | $0.009 \pm 0.0003$ | $0.011 \pm 0.0005$
>
> This clearly demonstrates that our non-amortized strategy yields highly competitive performance, effectively mitigating the amortization gap. Moreover, our method retains computational efficiency by limiting the optimization to only $L$ parameters per instance.
>
> **Comparison against sparse GP approx.** Thank you for giving us the opportunity to clarify also this important aspect. The LVAE and SVGP-VAE methods both use sparse GP approximation. LVAE method also assumes the same additive GP prior structure as our proposed model, whereas SVGP-VAE is limited to object-view product kernel. In other words, LVAE and SGP-VAE models that are already included in our comparisons implement exactly the kind of comparison that the reviewer is asking. In all experiments that we conducted, we provided a comparison of our method that leverages Hilbert space approximation to those that use sparse GP approximations. We will clarify this in the revised manuscript.

---

### Official Review · Reviewer_VgcQ · 2025-03-13

**Overall Recommendation:** 3

**Summary:**

This paper proposes a generative model based on Variational Autoencoders (VAE) where latent variables are assigned a GP prior.
The main contribution is to approximate the GP prior with random features so that the model can be optimized through mini-batching and linearly in the number of data.

**Claims And Evidence:**

The claim is that this approach offers better and more interpretable modeling compared to other competitors from the literature.
There are some experiments showing better performance but I couldn't find a thorough analysis on interpretability; I believe that this could be a major selling point of the paper and I would encourage the Authors to focus on this aspect more.

**Essential References Not Discussed:**

There is an ICML 2023 paper presenting a Bayesian autoencoder where latent variables are given a sparse GP (and Deep GP) prior, which I think could be a good competitor for this work:

[1] B.-H. Tran, B. Shahbaba, S. Mandt, and M. Filippone. Fully Bayesian Autoencoders with Latent Sparse Gaussian Processes. ICML 2023.

**Experimental Designs Or Analyses:**

Due to the experimental nature of the paper, I think that the experimental campaign needs to be extensive.
Overall, I think that the Authors did a good job in selecting a wide range of data sets and reporting some performance measures against competitors.
Again, I would encourage the Authors to focus their efforts on the intepretation of the results; this seems to be one of the main selling points of the proposed parameterization fo the GP prior and I believe it needs to be expanded on.

**Methods And Evaluation Criteria:**

I found the evaluation appropriate and the number of benchmark datasets provided to be sufficient.

**Other Comments Or Suggestions:**

Overall the paper is well written, so I don't have specific suggestions for changes in the the writing.

**Other Strengths And Weaknesses:**

I think that one of the main contributions is not highlighted too well in the paper, that is the one about parameterization.
The paper mentioned that thanks to the approximation and inference strategy there is no need for amortized inference; I believe this point could be emphasized in the main text (maybe by dedicating some space to this in Sec 4?) and in the experiments (e.g., by showing the advantage of no amortized inference vs the proposed parameterization).

Overall, one weakness is that the paper risks being just another model in the GP-VAE "zoo"; random feature approximations for GPs and Deep GPs are rather common and this might give the impression that the paper is a straightforward combination of known elements.
I would encourage the Authors to think of ways in which their work can be clearly differentiated from others in this literature and focus any additional experimental efforts in showing the advantages associated with their proposal.

**Questions For Authors:**

I've mentioned a few aspects which could improve the paper, and I hope that the rebuttal period will be useful for the Authors to come up with some constructive arguments in favor of acceptance.

**Relation To Broader Scientific Literature:**

I think that the paper does a good job in characterizing the literature, with some exceptions (see below).

**Theoretical Claims:**

There are no theoretical developments in the paper.

---

> ### Author Rebuttal · Authors · 2025-03-31
>
> We appreciate the reviewer's feedback and the opportunity to clarify our contributions. Before addressing specific comments, we emphasize that, unlike traditional methods that use random feature approximations, our approach leverages the Hilbert space approximation of the GP prior.
>
> **Interpretation of the results.** We agree that the interpretability of our parameterization is important. Although quantifying interpretability with neural networks is inherently challenging, our latent function visualizations for the Health MNIST data (see Figure 1) offer clear qualitative insights into the model's behavior. In the final version, we will include additional figures and detailed discussions to further illustrate these insights.
>
> Please refer to our response to reviewer Fbnr (Q4) for an additional Sobol index metric that quantifies interpretability by measuring the contributions of each additive component.
>
> **Global parameterization.** We thank the reviewer for highlighting our elimination of amortized inference and the related amortization gap [1,2,3]. We will emphasize this contribution in the revised version and add experiments demonstrating the benefits of our approach over an amortized variant. In the DGBFGP model, the instance-specific additive latent components correspond to local latent variables that could be amortized. As detailed in Appendix (Section B.1), these components are modeled using a categorical kernel based on each individual’s "id". In the Hilbert space approximation, the instance-specific component becomes an additive offset in the latent space with a standard Gaussian prior, here denoted for simplicity as $\boldsymbol{a_p}$ for individual $p$. An amortized version then uses $q_{\phi}(\boldsymbol{a}_p \mid Y_p)$ as an amortized variational approximation of the true posterior, where $\phi$ denotes inference network that is *shared* across all individuals. Our ablation study shows that sharing an inference network across individuals leads to poorer performance compared to our proposed parameterization:
>
> |Method|Rotated MNIST|Health MNIST|
> |-|-|-|
> |Amortized DGBFGP|$0.010 \pm 0.0004$|$0.012 \pm 0.0002$|
> |DGBFGP|$0.009 \pm 0.0001$|$0.009 \pm 0.0000$|
>
> Simulated datasets using MNIST images allow for easy and accurate amortized inference since the unrotated digit images are available for each individual. However, for more complex datasets, amortized inference would require more complex networks, and the amortization gap is likely to increase. We will include these analyses, along with studies on other datasets, in the revised version.
>
> **Contributions.** GPPVAE [4] was the first GP prior VAE model, with subsequent works building incrementally on it; [5] infers GPs for each trajectory separately, while [6] and [7] use inducing points (with [7] focusing on longitudinal designs using additive kernels). Our contributions are as follows: 1) We argue that the utilization of inducing points in the presence of categorical covariates is problematic and a non-trivial task since their locations cannot be optimized with gradients. To our knowledge, no efficient general solution has been proposed to handle discrete covariates in latent variable models, GP prior VAEs in particular, so far. Our work solves those problems by using the Hilbert space approximation method. 2) All previous works need to explicitly handle the kernels and their approximations one way or the other. Our approach provides a direct way to avoid explicit kernels. 3) We also propose to handle kernel hyperparameters probabilistically using VI (we note that was also done in Tran et al [8]). 4) Instead of performing amortized inference using a shared inference network, we directly optimize the global parameters (provided by our approximation method). This avoids the amortization gap and thereby improves the performance. We now also show and quantify the amortization gap via additional experiments. 5) Overall, we demonstrate that our model is more scalable and outperforms previously proposed methods. 6) Additionally, motivated by reviewers' comments, we now better demonstrate the interpretability of the proposed model by visualizing the latent effects as well as quantifying the contributions of different effects (see our response to Q4 of the reviewer Fbnr).
>
> **Additional competitor.** We thank the reviewer for bringing up SGP-BAE [8]. We run this model on the synthetic datasets and we will include results for the remaining experiments in the final version.
>
> |Method|Rotated MNIST|Health MNIST|
> |-|-|-|
> |SGP-BAE|$0.023 \pm 0.0006$|$0.024 \pm 0.0077$|
> |DGBFGP|$0.009 \pm 0.0001$|$0.009 \pm 0.0000$|
>
> [1] https://proceedings.mlr.press/v80/cremer18a.html
>
> [2] https://proceedings.mlr.press/v84/krishnan18a.html
>
> [3] https://proceedings.mlr.press/v80/marino18a
>
> [4] https://arxiv.org/abs/1810.11738
>
> [5] https://arxiv.org/abs/1907.04155
>
> [6] https://arxiv.org/abs/2010.13472
>
> [7] https://arxiv.org/abs/2006.09763
>
> [8] https://arxiv.org/abs/2302.04534

---

### Official Review · Reviewer_Fbnr · 2025-03-13

**Overall Recommendation:** 3

**Summary:**

This work presents a scalable basis function-based approximation for Gaussian Process prior Variational Auto-Encoders (GP-VAEs), to overcome the cubic time-complexity (without resorting to inducing-point GP variational inference techniques) and to accomodate shared and individual-specific correlations across time.

Their method allows for continuous and categorical covariate information to be incorporated for conditional generation, due to the proposed Hilbert space kernel approximation based on the kernels' eigenvalue and eigenfunction decomposition (described in Section 4). More precisely, the authors propose an additive GP prior for VAEs, which is defined using such Hilbert space eigen-decomposition of kernels.

Due to the proposed decomposition, learning can be posed as a variational inference over global parameters (kernel hyperparameters and linear model parameters $A$), with run-time complexity that scales linearly in the size of the dataset (amenable to mini-batching).

Results are reported on synthetic and real-world datasets, showcasing good predictive performance (as measured by Mean Squared Error), at reduced computational complexity.

# After rebuttal
The authors provided informative clarifications and additional Sobol Index based analysis of their results for further illustration. I hence lean towards acceptance of the work, with revised final manuscript.

**Claims And Evidence:**

The claims are generally well-supported.

The core idea of using kernel eigen-decomposition for an scalable GP approximation is well-grounded in already established theory.

The paper effectively demonstrates the practical benefits of this approach through empirical evaluation.

**Essential References Not Discussed:**

The main references to related work are well described.

**Experimental Designs Or Analyses:**

The experimental setup, using modified benchmark datasets (MNIST, Physionet, SPRITES), is appropriate for demonstrating the method's ability to capture time-varying latent forces driving multi-dimensional time-series observations.

However, solely relying on the MSE as a metric for evaluation, limits the experimental assessment. Evaluating the quality of the learned latent time series, and its dependence over the number of eigen-functions $M$ used to approximate a kernel, would provide a more comprehensive picture.

**Methods And Evaluation Criteria:**

The proposed method, i.e., utilizing additive GP priors and kernel eigen-decomposition, is a sound alternative for GP-VAE inference.

The evaluation, primarily focused on MSE for predictive performance, is relevant to the task of time-series prediction.

However, expanding the evaluation to include metrics assessing the quality of the learned latent spaces would strengthen the analysis.

**Other Comments Or Suggestions:**

N/A

**Other Strengths And Weaknesses:**

Strengths:

- The key strength is the effective use of kernel eigen-decomposition to achieve linear-time complexity in GP-VAE inference.
- The ability to handle both continuous and categorical covariates is a valuable contribution.

Weaknesses:

- The assumption of independence across latent dimensions and the use of additive GP priors per dimension could be limiting.

- The computational cost of computing the eigen-decomposition, especially for varying kernel hyperparameters, is not thoroughly discussed.

- The paper mainly focuses on Squared-Exponential kernels.

**Questions For Authors:**

- Please provide a detailed explanation of the computational procedure for calculating the eigen-decomposition of Squared-Exponential kernels, particularly on how this process scales with varying kernel hyperparameters. Quantifying the computational cost would be very helpful.

- Beyond the Squared-Exponential kernel, what other continuous kernels can be efficiently approximated using the proposed eigen-decomposition method?

- Could you elaborate on the challenges and potential solutions for extending the method to non-additive GP priors, specifically addressing the complexity of handling a non-fully factorized matrix A in Equation (4)?

- Would it be feasible to include an evaluation of the quality of the learned latent GP functions, particularly in synthetic experiments? Additionally, please discuss the sensitivity of the results to the choice of M, the number of eigenvalues and eigenfunctions used.

**Relation To Broader Scientific Literature:**

The key contribution of this work is to combine known results and ideas (eigen-decomposition of kernels with GP priors for VAEs) to propose a model variant that is computation efficient (linear complexity) and performant.

The authors provide a good overview and description of the main relevant literature (Sections 2, 3, 4.1-4.3), clearly explaining their novelty in combining kernel eigen-decomposition with an additive GP prior for VAEs, which can be defined via a global parametrization that implies increased computational efficiency.

**Theoretical Claims:**

The main theoretical results are:

1. The eigen-decomposition of kernels in Sections 4.1-4.3: these results are based on previously known results, so they are correct to the best of my knowledge.

2. The variational ELBO of Equation 7, with details in Appendix C: the presented expression appear correct upon review.

---

> ### Author Rebuttal · Authors · 2025-03-31
>
> **Independence across latent dims and additive GP prior** Correlated GP priors are typically formulated via the linear model of co-regionalization (LMC), which multiplies independent GPs by a factor loading matrix to introduce correlations across latent dimensions. In GP prior VAE models, a neural network–parameterized decoder maps latent variables to likelihood parameters, automatically introducing correlations at least as expressive as those from LMC. This approach is standard in previous GP prior VAE models, so assuming independence across dimensions does not lose generality or cause any limitation.
>
> Using additive GPs allows the latent effects to be decomposed into additive terms while leveraging a scalable basis function formulation. Although non-additive kernels may be more expressive in theory, our results show that DGBFGP significantly outperforms previous methods—even some that employ non-additive kernels.
>
> **Answer for Q1.** The Hilbert space approximation for GPs has a key property: the eigendecomposition of the Laplace operator with Dirichlet conditions is available in closed form and is independent of the kernel [1]. Similarly, a stationary kernel’s Hilbert space approximation depends on hyperparameters only through its spectral density, which is known for common kernels like SE and Matern. Thus, eigen-decomposition computation is unnecessary; the eigenfunctions and eigenvalues can be used in a plug-and-play fashion with an overall cost of $O(1)$, even when hyperparameters vary. We treat hyperparameters as random variables (Section 4.2) and infer them using VI (Section 4.4).
>
> For a categorical covariate, we compute the eigendecomposition of a $C$ x $C$ matrix ($C$ is the number of categories), and for the discrete kernels used here, closed-form solution again yields a cost of $O(1)$. Moreover, for arbitrary categorical kernels, with varying or fixed parameters, computational overhead is negligible since $C$ is typically much smaller than $N$.
>
> We will clarify these details in the revised manuscript.
>
> [1] https://link.springer.com/article/10.1007/s11222-019-09886-w
>
> **Answer for Q2.** We focused on the SE kernel because it is probably the most commonly used and it also worked well in our experiments. In practice, the choice of the kernel depends on the application. As noted at the end of Section 4.1, the Hilbert space approximation applies to any stationary continuous covariance function with a known spectral density, including the common SE, Matern, and many other kernels found in the literature.
>
> **Answer for Q3.** The Hilbert space approximation applies to additive, product, and sums of product kernels (as in our work). For instance, an SE kernel that depends on all input covariates—with a distinct length scale per covariate is equivalent to a product of SE kernels, leading to a product of basis function approximations (as in Section 4.3). Here, the number of terms scales as $M^q$, with $M$ basis functions per covariate and $q$ covariates. This is computationally feasible for a small $q$ but not scalable in general. However, Eq. (4) still factorizes as $A \mid \boldsymbol{\sigma}, \boldsymbol{\ell} \sim \prod_{l = 1}^L \mathcal{N}(\boldsymbol{a}_l \mid \boldsymbol{0}, S(\sigma, \ell))$, where $S(\sigma, \ell)$ is again diagonal, meaning the Gaussian distributions become $M^q$ dimensional. A similar factorization holds even for dependent GP priors before applying LMC (see our response above), in which case modeling a separate factor loading matrix would introduce latent correlations (or would be incorporated into the decoder function as described above).
>
> **Answer for Q4.** Since assessing latent function *quality* is challenging, we propose to quantify the interpretability by measuring contributions of each additive component using the Sobol index, defined for component $r$ as $\frac{\mathrm{Var}[f^{(r)}(x^{(r)})]}{\mathrm{Var}[\sum_rf^{(r)}(x^{(r)})]}$ . The Sobol index values for the four datasets are shown below.
>
> |Dataset|Component|%|
> |-|-|-|
> |Rotated MNIST|id|48|
> ||rotation|52|
> |Health MNIST|age|3|
> ||id|71|
> ||age x gender|11|
> ||diseaseAge x disease|15|
> |Physionet|id|85|
> ||time|2|
> ||time x ICU|7|
> ||time x gender|2|
> ||time x mortality|4|
> |Sprites|time|1|
> ||time x body|4|
> ||time x bottom|4|
> ||time x top|11|
> ||time x hair|18|
> ||time x action|30|
> ||time x direction|32|
>
> The id component’s dominance is expected since it captures each instance’s primary distinguishing features and serves as a natural baseline. Other factors add nuance and boost model capacity.
>
> For visual interpretability, the latent functions in Figure 1 (from the Health MNIST experiment in Section 5.2) highlight the model’s interpretability. We will include additional visualizations and expand the discussion in the revised Appendix.
>
> Finally, the sensitivity to $M$ is reported in Appendix G (Figures 5 and 7), showing that DGBFGP achieves the best results with $M = 6$ for Rotated MNIST and $M = 4$ for Health MNIST.

---

> > ### Comment · Reviewer_Fbnr · 2025-04-03
> >
> > Dear authors,
> >
> > Thank you very much for your informative response.
> >
> > It is now clear that the eigendecomposition of the Laplace operator with Dirichlet conditions is available (independently of the used kernel) in closed form, and hence, for a given kernel, the computational cost is of order O(1).
> >
> > Thank you for the clarifications on why per-component, additive GPs make sense in the context of GP-VAEs, as well as your interpretability results using Sobol Indexes.

---

### Decision · Program_Chairs · 2025-05-01

**Decision:**

Accept (poster)

**Comment:**

The authors provided a strong rebuttal to the reviewers' critiques. In particular, the authors gave a convincing answer to the novelty of their method compared to other GP random features approximation methods. The experimental results also further strengthened their paper. Therefore, I vote to accept.